# AD-BTS: Adaptive Dual-Branch Token Sparsification via Spatial Information Density

**Xinpei Gao** [* 1 2]   **Xin Luo** [* 1 2]   **Ming Liu** [1 2]   **Chunjiang Wang** [1 2]   **S Kevin Zhou** [1 2 3 4 5]

## Abstract

High-resolution visual encoders in multimodal large language models (MLLMs) substantially improve fine-grained perception, yet incur prohibitive computational costs. Existing token pruning methods are effective on natural images but struggle with spatially sparse structured inputs (e.g., charts), where critical high-frequency information is sparse, localized, and structurally essential. To address this challenge, we propose Adaptive Dual-Branch Token Sparsification (**AD-BTS**), a density-aware framework that dynamically allocates computation according to input signal characteristics. Specifically, AD-BTS introduces a Gradient-based Routing Gate (GRG) that uses lightweight pixel-level gradient statistics to estimate structural flatness and guide routing. Then, AD-BTS activates either a Redundancy Selection Branch (RSB) for aggressive token pruning with a frozen encoder, or a Structural Fusion Branch (SFB) with conditional LoRA and context fusion to preserve sparse structural information. Extensive experiments on Qwen2.5-VL demonstrate that AD-BTS establishes a new Pareto frontier between efficiency and accuracy. Under extreme compression (20% token retention), AD-BTS outperforms the strongest baseline by 12.1% on ChartQA while achieving a 1.8× prefill speedup, effectively reconciling computational efficiency with structural robustness.

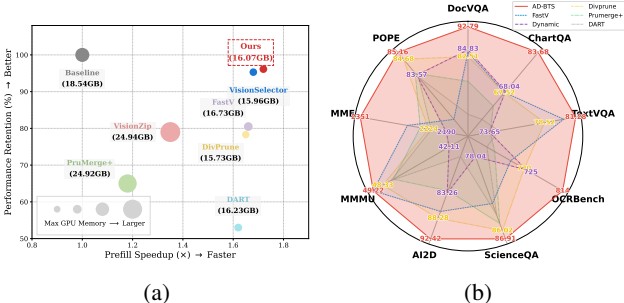

*Figure 1.* **Motivation and empirical evidence for adaptive token sparsification.** (a) Under a fixed 20% token budget, static sparsification methods exhibit a clear efficiency–accuracy trade-off on DocVQA, while AD-BTS achieves superior retention with lower memory cost. (b) On structure-sensitive benchmarks, AD-BTS consistently dominates prior methods under a 30% token budget, indicating the necessity of structure-aware computation allocation.

## 1. Introduction

The evolution of Multimodal Large Language Models (MLLMs) has shifted from coarse semantic alignment toward perceiving fine-grained visual details at higher spatial resolution (Radford et al., 2021; Jia et al., 2021). Recent architectures, such as LLaVA-NeXT (Liu et al., 2024) and Qwen2-VL (Wang et al., 2024), therefore universally adopt high-resolution visual encoders. While this design substantially improves visual fidelity, it introduces a severe token explosion problem: a single image can generate thousands of visual tokens, resulting in quadratic computational and memory overhead during the prefill stage of attention (Vaswani et al., 2017; Chen et al., 2024). This bottleneck critically limits the scalability of MLLMs in long-context video understanding and real-time interactive settings (Lin et al., 2023; Zhang et al., 2024; Chu et al., 2023).

Visual token sparsification has emerged as a promising approach to alleviate this inefficiency in large-scale multimodal systems under increasing visual resolution demands. Existing methods primarily rely on token pruning (Chen et al., 2024; Rao et al., 2021) or token merging (Bolya et al., 2023). Despite architectural differences, these approaches share a common implicit assumption: visual information is spatially redundant and background tokens can be safely

---

[*]Equal contribution  [1]School of Biomedical Engineering, Division of Life Sciences and Medicine, University of Science and Technology of China (USTC), China [2]Medical Imaging, Robotics, Analytic Computing & Learning (MIRACLE) Lab, YRD-RIGHT, USTC Suzhou Institute for Advanced Research, Suzhou, China [3]State Key Laboratory of Precision and Intelligent Chemistry, USTC, China [4]Jiangsu Provincial Key Laboratory of Multimodal Digital Twin Technology, USTC, China [5]Biomedical Basic Research Center (BBRC) of Jiangsu Province, Suzhou, China. Correspondence to: S Kevin Zhou <skevinzhou@ustc.edu.cn>.

*Proceedings of the 43rd International Conference on Machine Learning*, Seoul, South Korea. PMLR 306, 2026. Copyright 2026 by the author(s).

discarded (Liang et al., 2022). While this assumption generally holds for natural images with smooth spatial correlations, it breaks down for spatially sparse structured inputs such as charts and documents (Liu et al., 2022). In such cases, semantic information is concentrated in sparse, high-frequency structures (e.g., lines, glyphs, grid intersections) surrounded by large low-information regions. Applying uniform redundancy-based sparsification to these inputs risks fracturing the underlying structural skeleton, leading to disproportionate semantic degradation.

We argue that effective token sparsification must move beyond static, one-size-fits-all operators and instead adapt to input signal characteristics. A static sparsifier cannot simultaneously achieve aggressive compression for redundancy-dominated inputs while preserving fragile structures in sparse symbolic data (Han et al., 2015; Molchanov et al., 2019). Rather than introducing heavyweight semantic routers, we seek a lightweight physical proxy that captures information distribution at minimal cost (Mu & Lin, 2025). To this end, we identify pixel-level gradient statistics as a strong inductive bias. Natural images typically exhibit dense gradient distributions, whereas structured symbolic inputs are characterized by structural flatness, where gradients are sparse and information is both localized and fragile. This observation motivates a density-aware decoupling strategy that prioritizes efficiency through pruning when redundancy dominates and activates compensatory mechanisms when structural sparsity is detected.

As empirically illustrated in Figure 1, static sparsification methods exhibit inconsistent degradation across tasks, with particularly severe performance loss on structure-sensitive benchmarks. This observation highlights the limitation of one-size-fits-all sparsifiers and motivates the need for input-adaptive computation. Guided by this insight, we propose Adaptive Dual-Branch Token Sparsification (AD-BTS), an efficiency-first inference framework that dynamically allocates computation based on input structure. At the pipeline entrance, a lightweight Gradient-based Routing Gate (GRG) estimates information density from pixel-level gradient statistics and identifies inputs with high structural flatness. Based on this routing decision, AD-BTS decouples processing into two complementary branches: a Redundancy Selection Branch (RSB) that aggressively prunes tokens for efficiency on redundancy-dominated inputs, and a Structural Fusion Branch (SFB) that preserves fragile structural information through context fusion and conditional Low-Rank Adaptation (LoRA) under extreme compression.

Our contributions are three-fold:

- **Density-Aware Dual-Branch Sparsification.** We propose AD-BTS, which reimagines token sparsification as dynamic resource allocation. By orchestrating a frozen Redundancy Selection Branch and a compensatory Structural Fusion Branch, we effectively decouple the conflicting goals of efficiency and structural integrity.

- **Lightweight Gradient-Based Routing.** We introduce the Gradient-based Routing Gate, a parameter-free routing mechanism that identifies structure-sensitive inputs from pixel-level gradient statistics.

- **Performance Breakthrough at High Compression.** AD-BTS establishes a new efficiency–accuracy Pareto frontier, outperforming prior methods on structure-sensitive benchmarks under 10–30% token retention while achieving substantial prefill speedups.

**Conflict of Interest Disclosure** The authors declare no competing interests.

## 2. Related Work

### 2.1. High-Resolution MLLMs and Token Explosion

The trajectory of Multimodal Large Language Models (MLLMs) has evolved from aligning global semantic features (Radford et al., 2021; Jia et al., 2021) to perceiving fine-grained visual details. Early architectures projected fixed-resolution image embeddings into the LLM space, often inducing hallucinations in detail-sensitive tasks such as OCR (He et al., 2025). To address this limitation, recent models adopt high-resolution and dynamic resolution strategies. LLaVA-NeXT (Liu et al., 2024) employs an "AnyRes" mechanism to preserve aspect ratios through grid partitioning, while Qwen2-VL (Wang et al., 2024) introduces M-RoPE to process images at native scales.

While these designs substantially improve perceptual fidelity, they introduce a severe token explosion problem. A single high-resolution image can generate thousands of visual tokens, leading to quadratic computational complexity and memory overhead in the KV cache during the prefill stage (Wan et al., 2023). This bottleneck motivates the need for token reduction strategies that not only alleviate computational cost but also adapt to the heterogeneous information density inherent in high-resolution visual inputs.

### 2.2. Visual Token Adaptive Compression

**Selection-based methods (Pruning)** rely on the premise of spatial redundancy. Approaches like DynamicViT (Rao et al., 2021) employ learnable predictors to hierarchically discard uninformative tokens, while FastV (Chen et al., 2024) exploits the "attention bottleneck" in deep layers to prune tokens without retraining. While effective for redundancy-heavy images, aggressive pruning creates structural vulnerabilities, risking loss of high-frequency details in spatially sparse inputs and disrupting the positional grid.

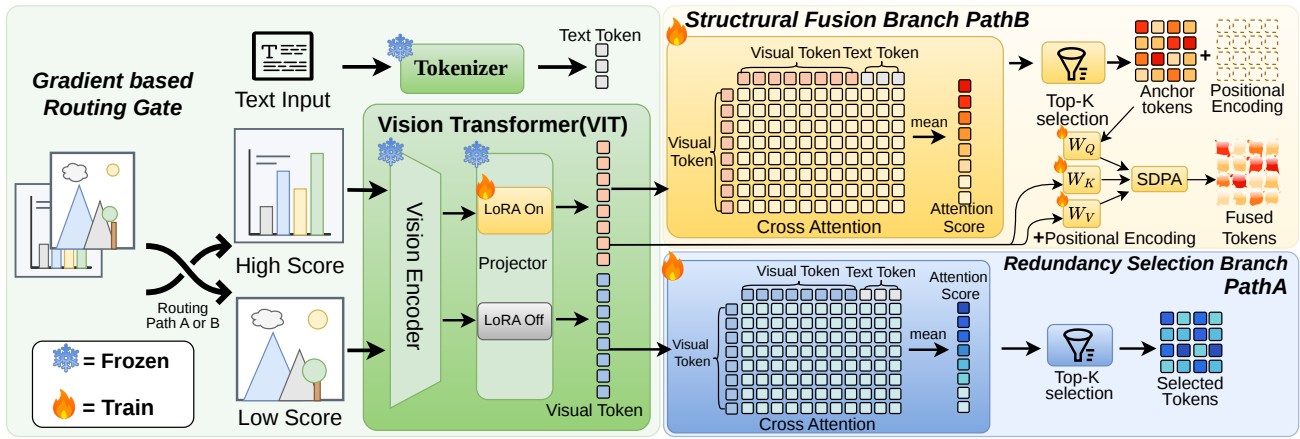

*Figure 2.* **Overview of the Adaptive Dual-Branch Token Sparsification via Spatial Information Density(AD-BTS) framework.** The architecture consists of three key components: (1) The Gradient-based Routing Gate (GRG) estimates spatial information density (sparsity) to route inputs and configure Dynamic Adapter Modulation (DAM). (2) Path A: Redundancy Selection Branch (RSB) processes redundancy-dominated inputs by keeping LoRA frozen (**Frozen**) and performing hard token selection. (3) Path B: Structural Fusion Branch (SFB) handles spatially sparse structured inputs by enabling LoRA adapters (**Train**) and fusing context into anchor tokens via FlashAttention-based cross-attention. The dynamic routing ensures efficient inference without sacrificing capacity for structural data.

**Fusion-based methods (Merging)** seek to preserve information through aggregation. Token Merging (ToMe) (Bolya et al., 2023) utilizes bipartite soft matching to gradually combine similar tokens, employing proportional attention to maintain feature magnitude. VisionZip (Yang et al., 2024) advances this via an encoder-guided strategy. Rather than relying on cross-attention for compression, it identifies dominant tokens via self-attention scores within the vision encoder itself, keeping salient tokens intact while merging the remaining context to minimize redundancy.

**Hybrid and Adaptive Paradigms** attempt to transcend static trade-offs. Frameworks like ToFu (Kim et al., 2023) introduce adaptability based on functional linearity: applying pruning in early layers where linearity is low, and switching to merging in deeper layers. However, these methods typically adapt based on internal network properties rather than the intrinsic information density of the input. **In contrast**, AD-BTS utilizes pixel-level gradient statistics to decouple computation: applying aggressive selection for redundancy-heavy inputs while switching to structure-preserving fusion for spatially sparse structured data.

Meanwhile, dynamic computation paradigms (e.g., MoE (Mu & Lin, 2025)) and parameter-efficient fine-tuning methods (e.g., LoRA (Hu et al., 2022), MoLE (Jie et al., 2025)) seek to adjust model capacity. However, these approaches usually activate adapters persistently and operate independently of token sparsification. **By contrast**, AD-BTS jointly considers token compression and adaptive computation. It conditions both the sparsification strategy and adapter activation on input structure, dynamically balancing efficiency and structural fidelity.

## 3. Method

### 3.1. Problem Formulation

Let $\mathbf{X} \in \mathbb{R}^{N \times D}$ denote the sequence of visual tokens projected from an input image $I$, and $\mathbf{T} \in \mathbb{R}^{M \times D}$ denote the corresponding text instruction embeddings. In a standard Multimodal Large Language Model (MLLM), the autoregressive generation conditions on the full concatenation $[\mathbf{X}; \mathbf{T}]$, leading to a computational complexity of $\mathcal{O}((N + M)^2)$. Our goal is to learn a dynamic sparsification operator $\Phi(\mathbf{X}, \mathbf{T}) \to \tilde{\mathbf{X}}$ where $\tilde{\mathbf{X}} \in \mathbb{R}^{K \times D}$ represents a selected subset of visual tokens with cardinality $K \ll N$, determined by a budget ratio $\rho$ such that $K = \lfloor \rho N \rfloor$. Crucially, this selection must be task-aware (guided by $\mathbf{T}$) and structure-aware (adaptive to image complexity).

### 3.2. The AD-BTS Framework

We propose **Adaptive Dual-Branch Token Sparsification (AD-BTS)**, a structure-aware framework that dynamically allocates computation based on input information density. As illustrated in Figure 2, AD-BTS follows a modular pipeline that decouples routing, selection, and execution. Specifically, a lightweight *Gradient-based Routing Gate (GRG)* first estimates spatial redundancy to determine the appropriate execution path. Conditioned on this decision, a task-aware *Differentiable Top-K* mechanism selects a compact subset of visual tokens for downstream processing. Finally, the selected tokens are processed by one of two specialized branches: the *Redundancy Selection Branch (RSB)* for efficiency-oriented pruning, or the *Structural Fusion Branch (SFB)* for structure-preserving computation.

## 3.3. Gradient-based Routing Gate

The GRG functions as a computationally negligible inductive bias to quantify structural flatness. Spatially sparse inputs are characterized by vast uniform backgrounds punctuated by high-frequency structural edges. We quantify this by analyzing the sparsity of the gradient field. Given the input image pixel values $I \in \mathbb{R}^{H \times W \times 3}$, we compute the gradient magnitude map using finite differences. We define a *Flatness Score* $\mathcal{S}_{\text{flat}}$ as the proportion of the spatial domain $\Omega$ where the local gradient magnitude is negligible:

$$\mathcal{S}_{\text{flat}}(I) = \frac{1}{|\Omega|} \sum_{(u,v) \in \Omega} \mathbb{I}\left(\|\nabla I_{u,v}\|_1 < \tau\right), \qquad (1)$$

Here, $\mathbb{I}(\cdot)$ is the indicator function, and $\nabla I_{u,v}$ approximates the pixel-level gradient magnitude via absolute differences $|\Delta_x| + |\Delta_y|$. With a noise threshold $\tau$, the resulting score $\mathcal{S}_{\text{flat}}(I)$ dictates the routing bifurcation: inputs are directed to the **RSB** if $\mathcal{S}_{\text{flat}}(I) \leq \delta$, and to the **SFB** if $\mathcal{S}_{\text{flat}}(I) > \delta$, where $\delta$ serves as the decision boundary.

## 3.4. Task-Aware Importance Scoring

For both branches, we employ a lightweight scoring module to evaluate the semantic relevance of each visual token. To ensure the selection is query-dependent, we formulate the scorer as a distinct attention mechanism where visual tokens query both themselves and the text tokens. Let $\mathbf{Q} = \mathbf{X}\mathbf{W}_Q$ and $\mathbf{K} = [\mathbf{X}; \mathbf{T}]\mathbf{W}_K$, where $[\cdot; \cdot]$ denotes concatenation along the sequence dimension. The importance score vector $\mathbf{s} \in \mathbb{R}^N$ is computed as:

$$\mathbf{A} = \frac{\mathbf{Q}\mathbf{K}^\top}{\sqrt{D}}, \quad \mathbf{s}_i = \frac{1}{N+M} \sum_{j=1}^{N+M} \mathbf{A}_{i,j}, \qquad (2)$$

where $\mathbf{W}_Q, \mathbf{W}_K$ are learnable projections initialized to yield near-zero variance, ensuring minimal disturbance at the start of training.

## 3.5. Differentiable Top-K Selection

Optimizing discrete selection (hard Top-K) is non-differentiable. We propose a differentiable relaxation based on implicit differentiation.

**Forward Pass (Root-Finding).** Instead of sorting, we seek a dynamic bias term $b^*$ that shifts the scores such that the sum of their sigmoid activations equals the target budget $K$. We solve the following root-finding problem via the bisection method:

$$\text{Find } b^* \in \mathbb{R} \quad \text{s.t.} \quad \sum_{i=1}^{N} \sigma(s_i + b^*) = K, \qquad (3)$$

where $\sigma(\cdot)$ is the sigmoid function. The soft mask is then defined as $\mathbf{m} = \sigma(\mathbf{s} + b^*)$. As training progresses, the magnitude of $\mathbf{s}$ increases, causing $\mathbf{m}$ to approximate a binary mask.

**Backward Pass (Implicit Gradients).** We strictly define the gradient flow for the custom autograd function. Let $\mathcal{L}$ be the task loss. The gradient with respect to the scores $\mathbf{s}$ is derived using the Implicit Function Theorem to account for the dependency of $b^*$ on $\mathbf{s}$:

$$\frac{\partial \mathcal{L}}{\partial \mathbf{s}} = \left( \frac{\partial \mathcal{L}}{\partial \mathbf{m}} - \frac{\langle \frac{\partial \mathcal{L}}{\partial \mathbf{m}}, \sigma'(\mathbf{z}) \rangle}{\|\sigma'(\mathbf{z})\|_1} \right) \odot \sigma'(\mathbf{z}), \qquad (4)$$

where $\mathbf{z} = \mathbf{s} + b^*$ and $\odot$ is the Hadamard product. This mechanism introduces global competition: increasing the score of one token implicitly suppresses others to maintain the cardinality constraint $K$.

## 3.6. Dual-Branch Execution via Dynamic Adapter

To reconcile the conflicting requirements of structural plasticity for spatially sparse structured inputs and feature preservation for spatially redundant inputs, we propose a unified execution framework governed by a binary modulation gate $g \in \{0, 1\}$. Derived from the flatness prior Eq. (1), the gate is defined as $g = \mathbb{I}(\mathcal{S}_{\text{flat}}(I) > \delta)$.

This gate controls the **Dynamic Adapter Modulation (DAM)** mechanism within the visual encoder. While standard LoRA updates weights via $\Delta \mathbf{W} = \frac{\alpha}{r}\mathbf{B}\mathbf{A}$, we condition this update on $g$. The effective forward pass for a linear layer $\ell$ becomes:

$$\mathbf{h} = \mathbf{W}_0\mathbf{x} + g \cdot \left( \frac{\alpha}{r} \mathbf{B}_\ell \mathbf{A}_\ell \mathbf{x} \right), \qquad (5)$$

where $\mathbf{W}_0$ is the frozen pretrained weight, $\mathbf{B}_\ell \in \mathbb{R}^{d \times r}$ and $\mathbf{A}_\ell \in \mathbb{R}^{r \times d}$ are trainable low-rank matrices with rank $r$, and $\alpha$ is a scaling factor. Based on the state of $g$, the framework bifurcates into two distinct execution paths:

**RSB: Select-and-Freeze ($g = 0$).** For inputs dominated by spatial redundancy, the gate $g = 0$ strictly disables the adapters, reducing Eq. (5) to $\mathbf{h} = \mathbf{W}_0\mathbf{x}$. This ensures the visual backbone remains frozen to prevent overfitting to the reduced token set. In this baseline path, we perform a query-aware hard selection based on the importance scores $\mathbf{s}$. The output is a direct subset of the original features: $\tilde{\mathbf{X}} = \{\mathbf{x}_i \mid i \in \text{top-}K(\mathbf{s})\}$.

**SFB: Fuse-and-Adapt ($g = 1$).** For spatially sparse structured inputs, the gate $g = 1$ activates the LoRA adapters, enabling subspace adaptation to capture high-frequency structural details. To prevent severing the "structural skeleton" during sparsification, we employ a Query-Aware Feature Fusion strategy. We treat the selected top-$K$ tokens as *anchors* and the full feature map as the *global context*. To

strictly preserve spatial layout, we inject learnable positional encodings $\mathbf{P}$ into both representations prior to fusion. Let $\mathbf{H}_{\text{anchor}} = \mathbf{X}_{\mathcal{I}} + \mathbf{P}_{\mathcal{I}}$ and $\mathbf{H}_{\text{context}} = \mathbf{X} + \mathbf{P}_{\text{all}}$ denote the position-aware inputs. The fused representation $\tilde{\mathbf{X}}$ is computed via Multi-Head Attention (MHA) followed by a residual connection:

$$\tilde{\mathbf{X}} = \text{LayerNorm}\left(\mathbf{X}_{\mathcal{I}} + \text{MHA}(\mathbf{H}_{\text{anchor}}, \mathbf{H}_{\text{context}}, \mathbf{H}_{\text{context}})\right). \quad (6)$$

This path ensures that while the token count is reduced, the semantic integrity of the visual architecture is preserved through both feature fusion and parameter adaptation.

### 3.7. Training Objective

The entire framework is trained end-to-end using a unified objective function, applied consistently across both RSB and SFB paths. The total loss $\mathcal{L}_{\text{total}}$ is a weighted sum of the language modeling loss and a budget regularization term:

$$\mathcal{L}_{\text{total}} = \mathcal{L}_{\text{CE}} + \lambda_{\text{reg}} \cdot \mathcal{L}_{\text{budget}}, \quad (7)$$

where $\lambda_{\text{reg}}$ is a hyperparameter that balances the trade-off between semantic fidelity and the sparsity constraint.

**Language Modeling Loss ($\mathcal{L}_{\text{CE}}$).** This is the standard cross-entropy loss over the generated text tokens $\mathbf{Y}$, conditioned on the sparsified visual context $\tilde{\mathbf{X}}$ and the text instruction $\mathbf{T}$:

$$\mathcal{L}_{\text{CE}} = -\sum_{t=1}^{|\mathbf{Y}|} \log P(y_t \mid y_{<t}, \tilde{\mathbf{X}}, \mathbf{T}). \quad (8)$$

**Budget Regularization ($\mathcal{L}_{\text{budget}}$).** To bridge the discrepancy between the differentiable soft mask used for gradient estimation and the discrete selection used during inference, we employ a self-distillation mechanism. Recall the soft mask $\mathbf{m} \in (0, 1)^N$ derived in Eq. (3) (Forward Pass). Let $\mathbf{m}_{\text{hard}} \in \{0, 1\}^N$ be the binary mask corresponding to the top-$K$ indices $\mathcal{I}$. We define the regularization term as the Binary Cross-Entropy (BCE) between them:

$$\mathcal{L}_{\text{budget}} = \text{BCE}(\mathbf{m}, \mathbf{m}_{\text{hard}}). \quad (9)$$

This term encourages the soft mask $\mathbf{m}$ to converge towards the binary decision boundary, stabilizing the root-finding process in Eq. (3) and ensuring that the gradients computed via Eq. (4) accurately reflect the token selection importance.

## 4. Experiments

### 4.1. Experimental Setup

**Implementation Details.** We implement AD-BTS on top of Qwen2.5-VL-7B (Bai et al., 2025), finetuning on a hybrid dataset comprising the training splits of ChartQA and OCR-VQA, augmented with a 10% random subset of COCO to maintain general capabilities. To ensure parameter efficiency, we freeze the entire backbone and only update the branch-specific scorers and LoRA modules ($r = 256$) in the SFB. This introduces merely $\approx$85M trainable parameters (a 1.2% relative parameter increase), ensuring minimal storage overhead ($\approx$170MB).

**Evaluation Protocol.** We rigorously benchmark AD-BTS against 7 state-of-the-art baselines (comprising 5 training-free and 2 training-based methods) across 9 image benchmarks (covering general perception, OCR, and charts) and 3 video datasets. All methods are evaluated under strictly controlled token budgets with retention ratios $\rho \in \{0.3, 0.2, 0.1\}$ on NVIDIA H20 GPUs. The routing threshold is set to $\delta = 0.4$ based on ablations. We refer readers to Appendix E for detailed dataset statistics and baseline configurations.

### 4.2. Main Results

**Overall Performance under Aggressive Compression.** Table 1 reports quantitative results under three compression regimes. Across all settings, AD-BTS consistently establishes a new state-of-the-art. At moderate compression (30% retention), AD-BTS preserves 98.72% of the uncompressed upper-bound performance, surpassing the strongest training-based baseline (VisionSelector, 96.90%) and the best training-free method (VisionZip, 93.57%). Crucially, as compression becomes more aggressive, the performance gap widens.At the extreme 10% retention setting, where most pruning and merging strategies degrade sharply, AD-BTS maintains a robust 88.43% accuracy.This represents a substantial margin over strictly pruning-based methods like FastV (75.35%) and DART (68.32%), validating our hypothesis that simple token removal is insufficient for high-compression scenarios where spatial information density varies significantly.

**Robustness across Signal Regimes.** Performance degradation is highly task-dependent, reflecting the distinction between spatially redundant inputs and spatially sparse inputs. On structure-sensitive benchmarks such as ChartQA, uniform sparsification methods suffer pronounced failures due to the loss of critical high-frequency signals. For instance, at 10% retention on ChartQA, VisionSelector drops to 61.6% accuracy, whereas AD-BTS maintains 68.24%. Similarly, on OCRBench, AD-BTS outperforms DART by over 245 points. These results confirm that the Structural Fusion Branch (SFB), empowered by conditional LoRA capacity, effectively mitigates the fracture of fine-grained details. Conversely, on redundancy-dominated benchmarks like ScienceQA and POPE, AD-BTS remains highly competitive. Notably, on POPE, AD-BTS achieves an F1 score

*Table 1.* **Comparison with state-of-the-art methods across varying token retention rates.** We evaluate methods on nine benchmarks under three compression regimes (30%, 20%, and 10% retained tokens). The table reports absolute metrics for individual tasks, while the *Avg Score* column presents the performance normalized relative to the uncompressed Qwen2.5-VL-7B upper bound. **Bold** highlights the best results among compressed models.

| Method | Training Free | DocVQA (Anls) | ChartQA (Relaxed) | TextVQA (EM) | OCRBench (Score) | ScienceQA (EM) | AI2D (Acc) | MMMU (Acc) | MME (Score) | POPE (F1) | Avg Score (Norm %) |
|---|---|---|---|---|---|---|---|---|---|---|---|
| *Dynamic Resolution (MinPix=256x28x28, MaxPix=2048x28x28), Upper Bound (100%)* | | | | | | | | | | | |
| Qwen-2.5-VL-7B | – | 94.33 | 83.4 | 82.84 | 838 | 87.26 | 93.59 | 50.78 | 2342.15 | 86.19 | 100.00% |
| *Retain 30% Tokens (70% Compression Ratio)* | | | | | | | | | | | |
| DART | ✓ | 62.6 | 56.88 | 74.45 | 629 | 84.33 | 75.94 | 47.56 | 2218.83 | 83.43 | 84.72% |
| Prumerge+ | ✓ | 73.95 | 62.24 | 73.71 | 648 | 85.23 | 82.77 | 47.33 | 2239.64 | 83.69 | 87.93% |
| FastV | ✓ | 84.01 | 67.64 | 80.22 | 687 | 83.09 | 86.92 | 49.22 | 2263.58 | 80.47 | 91.56% |
| Divprune | ✓ | 82.51 | 67.52 | 78.52 | 720 | 86.02 | 88.28 | 48.33 | 2224.06 | 84.68 | 92.27% |
| Visionzip | ✓ | 86.11 | 72.28 | 77.3 | 711 | 86.61 | 87.86 | 49.44 | 2276.04 | 84.73 | 93.57% |
| Dynamic | ✗ | 84.83 | 68.04 | 73.65 | 725 | 78.04 | 83.26 | 42.11 | 2190.78 | 83.57 | 88.75% |
| Visionselector | ✗ | 92.61 | 73.04 | **81.28** | 801 | 85.77 | 91.94 | **50.11** | 2313.74 | 85 | 96.90% |
| AD-BTS (Ours) | ✗ | **92.79** | **83.68** | **81.28** | **814** | 86.91 | 92.42 | 49.22 | **2351.76** | 85.16 | **98.72%** |
| *Retain 20% Tokens (80% Compression Ratio)* | | | | | | | | | | | |
| DART | ✓ | 73.81 | 57.88 | 73.86 | 648 | 84.33 | 82.29 | 46.33 | 2198.82 | 83.55 | 86.75% |
| Prumerge+ | ✓ | 50.03 | 47.16 | 67.53 | 537 | 83.34 | 71.05 | 46.33 | 2138.61 | 80.16 | 78.02% |
| FastV | ✓ | 74.75 | 62.04 | 72.03 | 591 | 84.68 | 82.32 | 47 | 2168.86 | 83.23 | 86.43% |
| Divprune | ✓ | 75.99 | 60.48 | 78.01 | 597 | 82.75 | 82.35 | **49** | 2152.74 | 76.12 | 86.45% |
| Visionzip | ✓ | 61.09 | 52.56 | 68.72 | 562 | 83.79 | 77.62 | 46.11 | 2219.3 | 81.74 | 81.91% |
| Dynamic | ✗ | 78.09 | 66.20 | 72.22 | 675 | 77.00 | 81.70 | 43.78 | 2105.20 | 81.95 | 86.29% |
| Visionselector | ✗ | 89.78 | 68.12 | 79.79 | 763 | 85.42 | 90.48 | 48.78 | **2273.94** | 84.29 | 94.42% |
| AD-BTS (Ours) | ✗ | **90.22** | **80.2** | **80.15** | **766** | 85.97 | 90.67 | 48.33 | 2258.88 | **84.3** | **96.09%** |
| *Retain 10% Tokens (90% Compression Ratio)* | | | | | | | | | | | |
| DART | ✓ | 33.13 | 34 | 53.97 | 415 | 81.85 | 67.1 | 46.11 | 1980.7 | 71.91 | 68.32% |
| Prumerge+ | ✓ | 42.08 | 41.56 | 56.87 | 417 | 81.56 | 71.08 | 45.22 | 1948.58 | 76.52 | 71.48% |
| FastV | ✓ | 58.64 | 44.64 | 70.83 | 440 | 81.95 | 75.74 | 45.78 | 1940.91 | 65.99 | 75.35% |
| Divprune | ✓ | 54.04 | 39.8 | 64.65 | 477 | 82.15 | 72.8 | 46 | 2015.66 | 79.27 | 75.61% |
| Visionzip | ✓ | 48.29 | 42.84 | 55.94 | 404 | 82.6 | 73.09 | 45.44 | 1944.04 | 78.46 | 72.73% |
| Dynamic | ✗ | 62.34 | 58.76 | 69.68 | 554 | 76.20 | 77.36 | 43.78 | 2023.10 | 76.35 | 79.77% |
| Visionselector | ✗ | 77.01 | 61.6 | **75.11** | 654 | **83.64** | 84.72 | 46.56 | **2146.97** | 79.53 | 87.36% |
| AD-BTS (Ours) | ✗ | **77.84** | **68.24** | 75 | **660** | 82.6 | **85.4** | **46.67** | 2134.74 | **80.42** | **88.43%** |

*Table 2.* **Efficiency and performance trade-off on video benchmarks.** We compare accuracy on MVBench, SEEDBench, and NextQA against memory usage and prefill latency. *oom* denotes out-of-memory. The best results among compressed methods are highlighted in **bold**.

| Method | MVBench Acc | SEEDBench Acc | NextQA WUPS | Max GPU mem (GB) | Prefill Time (ms) |
|---|---|---|---|---|---|
| Qwen2.5-VL-7B | 68.10 | 60.48 | 27.58 | 25.97 | 1413.34 |
| FastV | 65.75 | *oom* | 27.00 | 24.63 | 851.76 |
| DART | 65.80 | **61.00** | 26.84 | 18.93 | 832.58 |
| Divprune | 65.85 | 59.79 | 27.00 | 17.55 | 1184.00 |
| Visionselector | 66.55 | 59.82 | **27.10** | 17.57 | 760.82 |
| AD-BTS | **67.03** | 60.37 | **27.10** | **16.76** | **746.25** |

of 80.42 even at 10% retention, surpassing purely training-free methods like FastV (65.99%). This demonstrates that our Redundancy Selection Branch (RSB) successfully filters redundant tokens in spatially redundant inputs without compromisingthe pre-trained representation.

### 4.3. Efficiency and Computational Cost Analysis

**Pareto Efficiency and Task Sensitivity.** To rigorously quantify the trade-off between token reduction and task performance, Figure 3 illustrates the trade-off between performance retention and prefill speedup. AD-BTS consistently dominates the **Pareto frontier** across diverse benchmarks. This advantage is particularly pronounced on **structure-sensitive tasks** (e.g., ChartQA and OCRBench), where baselines like FastV and VisionSelector degrade rapidly as speedup exceeds $1.4\times$. In contrast, AD-BTS maintains high retention by protecting critical structural primitives via the Fusion Branch, offering the most efficient conversion of computational budget into accuracy.

**Scalability on Long-Context Video Tasks.** The efficiency gains of AD-BTS are critical in memory-constrained scenarios commonly encountered in practical deployments. Processing high-resolution video frames typically triggers memory spikes. As observed on SEEDBench (Table 2), pruning-based baselines like FastV fail to manage these spikes, resulting in out-of-memory (OOM) errors. In contrast, AD-BTS successfully completes inference by dynamically filtering temporal redundancy, capping peak memory at 16.76GB (a 35% reduction compared to the 25.97GB baseline). Furthermore, AD-BTS reduces prefill latency from 1413ms to 746ms (≈47% reduction), enabling real-time interaction in long-context video reasoning without

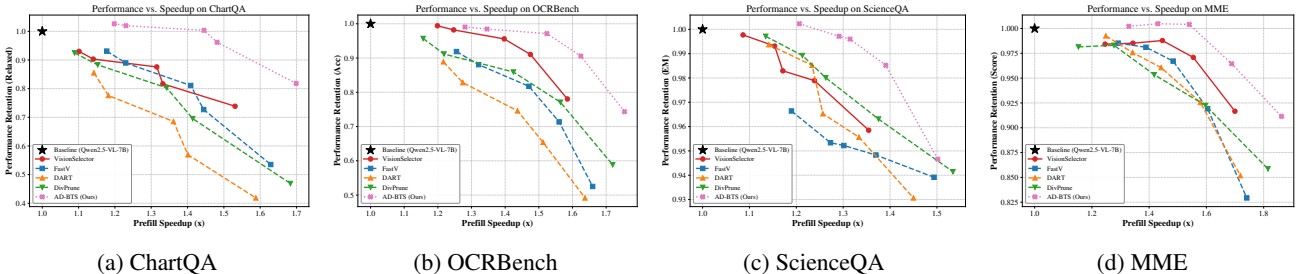

|  (a) ChartQA | (b) OCRBench | (c) ScienceQA | (d) MME |

*Figure 3.* **Performance Retention vs. Prefill Speedup.** Comparison of AD-BTS (pink dotted line) against state-of-the-art baselines. AD-BTS consistently dominates the Pareto frontier, particularly on sparse, structure-sensitive tasks (a, b), where baselines degrade rapidly as speedup increases.

*Table 3.* **Ablation Study on Hyperparameters.** We investigate the impact of the routing threshold $\delta$ and the LoRA rank $r$ on model performance. The upper section compares the dual-branch routing strategy against single-branch baselines ($\delta = 0$ for RSB-only, $\delta = 1$ for SFB-only). The lower section analyzes the sensitivity to LoRA capacity within the SFB path. **Bold** indicates the best performance among compressed variants (excluding the upper bound).

| Method | DocVQA (Anls) | ChartQA (Relaxed) | TextVQA (EM) | OCRBench (Score) | ScienceQA (EM) | AI2D (EM) | MMMU (Acc) | MME (Score) | POPE (F1) | Avg Score (Norm %) |
|---|---|---|---|---|---|---|---|---|---|---|
| *Upper Bound (100% Tokens)* | | | | | | | | | | |
| Qwen-2.5-VL-7B | 94.33 | 83.40 | 82.84 | 838 | 87.26 | 93.59 | 50.78 | 2342.15 | 86.19 | 100.00% |
| *Ablation on Routing Threshold $\delta$ (with Rank $r = 256$)* | | | | | | | | | | |
| $\delta = 0$ (RSB Only) | 92.90 | 73.12 | 81.31 | 802 | 86.22 | 91.97 | 49.56 | **2355.26** | 85.16 | 97.11% |
| $\delta = 0.2$ | 92.42 | **84.40** | **81.53** | 809 | 86.61 | 92.33 | 49.56 | 2308.04 | 85.16 | 98.55% |
| $\delta = 0.6$ | **92.92** | 74.68 | 81.31 | 805 | 86.66 | 92.13 | **50.00** | 2352.76 | 85.08 | 97.52% |
| $\delta = 1$ (SFB Only) | 92.46 | **84.40** | 81.39 | 812 | 86.66 | **92.42** | 48.67 | 2233.70 | 84.82 | 98.01% |
| *Ablation on SFB LoRA Rank $r$ (with $\delta = 0.4$)* | | | | | | | | | | |
| Rank $r = 0$ | 92.67 | 80.52 | 81.23 | **816** | 86.07 | 91.71 | 48.44 | 2339.18 | 85.07 | 97.86% |
| Rank $r = 64$ | 92.07 | 83.72 | 81.08 | 800 | 85.13 | 91.84 | 48.56 | 2231.66 | **85.80** | 97.50% |
| Rank $r = 128$ | 91.88 | 81.00 | 80.45 | 803 | 85.42 | 91.55 | 47.78 | 2237.61 | 84.88 | 96.81% |
| Rank $r = 512$ | 90.68 | 81.44 | 80.82 | 793 | 85.92 | 91.48 | 47.78 | 2254.00 | 85.24 | 96.82% |
| **AD-BTS**($r = 256, \delta = 0.4$) | 92.79 | 83.68 | 81.28 | 814 | **86.91** | **92.42** | 49.22 | 2351.76 | 85.16 | **98.72%** |

compromising temporal reasoning.

### 4.4. Ablation Study

We verify the effectiveness of our core design choices by conducting comprehensive ablation studies on the impact of routing threshold $\delta$ and the LoRA rank $r$ (Table 3).

**Impact of Dual-Branch Routing ($\delta$).** The threshold $\delta$ modulates the trade-off between the efficiency-oriented RSB path and the structure-preserving SFB path. RSB-Only ($\delta = 0$): Routing all inputs to the RSB yields the highest score on MME (2355.26), a benchmark dominated by spatially redundant inputs. However, performance significantly degrades on structure-sensitive tasks like ChartQA (73.12), confirming that simple selection is insufficient for complex layouts. SFB-Only ($\delta = 1$): Conversely, activating the fusion branch for all inputs recovers performance on ChartQA (84.40) and AI2D (92.42). Yet, this comes at the cost of lower scores on general perception tasks, suggesting that applying fusion to spatially redundant inputs may introduce unnecessary noise or "over-smoothing" of features. Opti-

mal density boundary ($\delta = 0.4$): Our proposed AD-BTS achieves the highest normalized average score (98.72%). The threshold $\delta = 0.4$ acts as a robust boundary that effectively separates safe-to-prune spatially redundant inputs from structure-sensitive sparse signals, retaining structural reasoning (ChartQA: 83.68) while maintaining general robustness (MME: 2351.76).

**Validation of Routing Distribution and Robustness.** To further validate that the flatness score $\mathcal{S}_{\text{flat}}$ physically separates structural information from redundancy, we evaluated the routing fractions across datasets using our optimal threshold ($\delta = 0.4$). As shown in Table 4, structure-sensitive benchmarks (e.g., ChartQA) are predominantly routed to the SFB (96.9%), while redundancy-heavy datasets (e.g., POPE) almost exclusively utilize the RSB (98.0%). This stark contrast proves the routing mechanism is highly discriminative.

Furthermore, the heuristic is inherently robust to misrouting. In cases of **False Positives** (e.g., a smooth sky image routed to SFB), our task-aware scorer still aggressively prunes irrel-

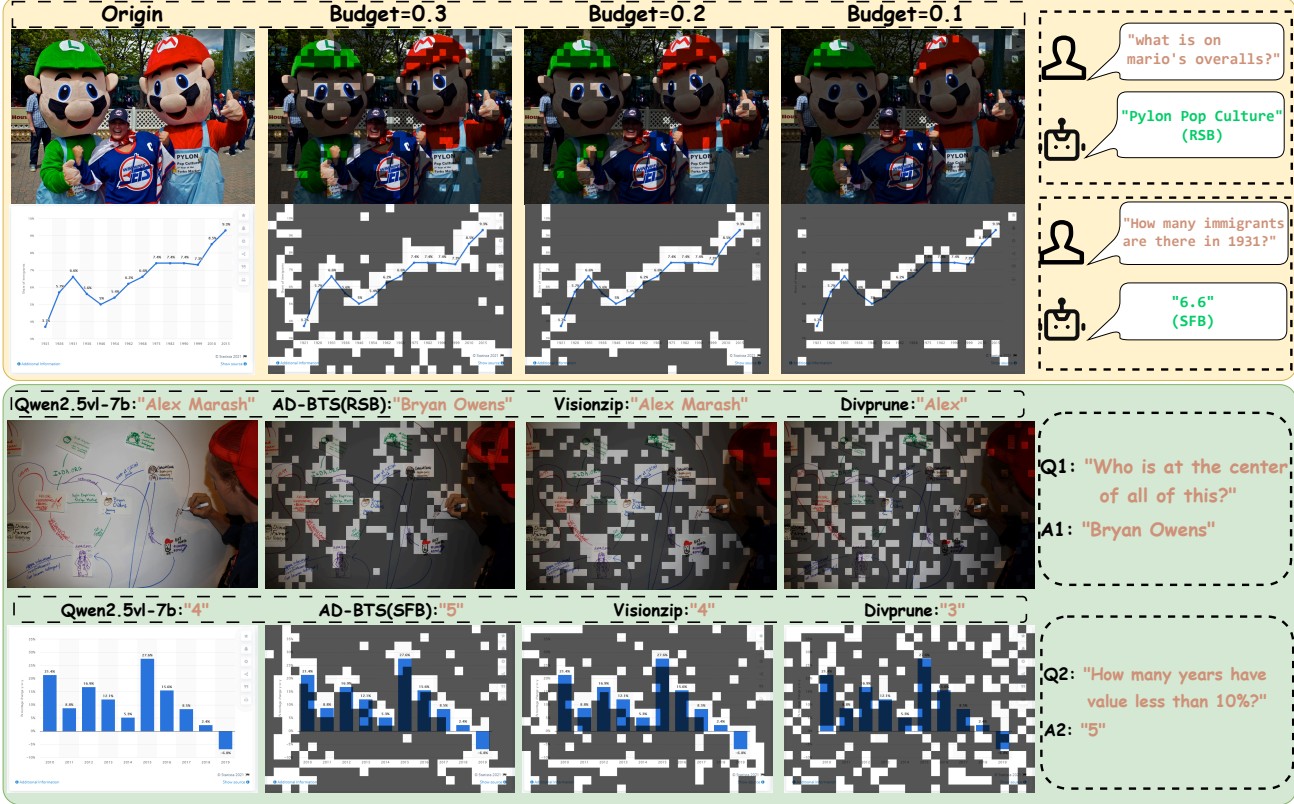

*Figure 4.* **Qualitative Visualization of Adaptive Token Retention. Top Panel (Budget Sensitivity):** (1) Under Spatial Redundancy (Natural Scene), the **RSB** eliminates background noise and selectively retains semantic regions guided by the query. (2) Under Spatial Sparsity (Chart), the **SFB** preserves the structural skeleton (e.g., trend lines, axes) via feature fusion, maintaining topology even at 10% retention. **Bottom Panel (Baseline Comparison):** Compared to VisionZip and DivPrune, AD-BTS avoids semantic distraction in natural images and prevents structural fragmentation in charts, yielding correct answers for both fine-grained reasoning and counting tasks.

evant background tokens before fusion; the only penalty is a negligible compute overhead (∼1.29 ms) without performance degradation. Conversely, for **False Negatives** (e.g., extremely dense text maps routed to RSB), retaining unmodified pretrained features via the RSB acts as a safe fallback for high-information-density scenarios. Thus, GRG serves as a fail-safe mechanism rather than a brittle condition.

*Table 4.* **Flatness-Score Distributions & Routing Fractions** ($\delta = 0.4$).

| Dataset | Mean $\mathcal{S}_{\text{flat}}$ | RSB (%) | SFB (%) |
|---|---|---|---|
| ChartQA | 0.546 | 3.1 | 96.9 |
| ScienceQA | 0.437 | 33.1 | 66.9 |
| OCRBench | 0.290 | 70.4 | 29.6 |
| MME | 0.214 | 90.4 | 9.6 |
| POPE | 0.156 | 98.0 | 2.0 |

**Impact of Adapter Capacity ($r$).** We further analyze the rank $r$ of the LoRA adapters within the SFB. Necessity of Learning ($r > 0$): Comparing $r = 0$ (frozen projection) with $r = 256$, we observe significant gains in ChartQA (+3.16%) and AI2D (+0.71%). This confirms that the pretrained visual manifold requires specific adaptation to recover information lost during pruning in spatially sparse

structured inputs. Capacity Trade-off: Performance peaks at $r = 256$. Lower ranks ($r = 64$) lack sufficient capacity to capture fine-grained structural details, while excessively high ranks ($r = 512$) lead to degradation (Avg Score 96.82%), likely due to overfitting the limited finetuning data.

**Source of Improvements: Mechanism vs. Capacity.** To strictly validate that the performance gains stem from our density-aware routing and fusion mechanisms rather than merely the addition of learnable parameters, we conduct controlled matched-capacity ablations (Table 5) at a 30% token budget ($\rho = 0.3$).

Specifically, we compare AD-BTS against two variants: (1) **Matched-Capacity (Always RSB + LoRA):** We force all inputs through the RSB (no context fusion) but persistently activate the LoRA adapters ($r = 256$) to match the parameter count of AD-BTS. This configuration fails to reconstruct fragile topological skeletons, causing significant degradation on structure-sensitive tasks (e.g., ChartQA drops from 83.68 to 78.64). This proves that simply expanding parameter capacity is insufficient without proper feature fusion. (2) **No-Fusion (Always SFB w/o Fusion):** We route inputs to the SFB and activate LoRA, but bypass the cross-attention

context fusion Eq. (6). This also results in suboptimal performance across all metrics.

Furthermore, as previously established in the routing ablation ($\delta = 1$), indiscriminately applying fusion to all inputs over-smooths features for natural images, dropping MME performance to 2233.70. These results jointly confirm that the dynamic routing and the structural fusion mechanisms are both indispensable and effectively complementary.

*Table 5.* **Controlled Ablations on Capacity and Fusion ($\rho =$ 0.3)**. We verify that performance gains derive from the proposed dual-branch mechanisms rather than merely increased parameter counts.

| Method / Configuration | ChartQA | DocVQA | MME | Avg (Norm) |
|---|---|---|---|---|
| Matched-Capacity (Always RSB + LoRA $r = 256$) | 78.64 | 92.00 | 2278.12 | 97.12% |
| No-Fusion (Always SFB w/o Eq. (6)) | 79.96 | 91.81 | 2255.70 | 96.81% |
| **AD-BTS (Ours)** | **83.68** | **92.79** | **2351.76** | **98.72%** |

### 4.5. Qualitative Analysis

**Effect of Budget Scaling (Top Panel of Fig. 4).** We first examine the retention behavior under varying sparsity levels $\rho \in 0.3, 0.2, 0.1$. For spatially continuous inputs (e.g., the mascot scene), the Gradient-based Routing Gate routes the input to the RSB. As the budget decreases, the model exhibits a coarse-to-fine pruning behavior. It retains the semantic foreground, including faces and text on clothing, while progressively filtering out the redundant crowd background. Conversely, for spatially sparse structured inputs, the SFB is activated. Even at an extreme compression rate of 90% ($\rho = 0.1$), AD-BTS preserves the discrete structural skeleton, including trend lines and axis ticks, which are crucial for reading data, whereas the information-sparse white background is almost entirely discarded.

**Comparison with SOTA Methods (Bottom Panel of Fig. 4).** We further compare AD-BTS against leading compression methods, VisionZip and DivPrune, to highlight qualitative differences in failure modes. In the reasoning task ("Who is at the center..."), baselines suffer from attentional misalignment, retaining irrelevant peripheral textures that distract the LLM (leading to incorrect answers like "Alex" or "Alex Marash"). In contrast, AD-BTS (RSB) utilizes query-aware scoring to localize the central actor ("Bryan Owens"), discarding clutter. In the fine-grained counting task ("How many years..."), baselines cause *structural fragmentation*, breaking the continuity of bars and axes required for numerical estimation. AD-BTS (SFB) maintains the geometric integrity of the bars and their alignment with the y-axis, enabling the model to correctly deduce the count "5".This robustness to structural decay proves essential for reliable quantitative reasoning in high-resolution document and chart analysis.

## 5. Conclusion

In this work, we identify and address a critical limitation in existing MLLM efficiency strategies: the static assumption of uniform spatial redundancy. We propose **AD-BTS**, a density-driven framework that utilizes gradient statistics as a lightweight physical proxy to dynamically decouple the processing of spatially redundant inputs from spatially sparse structured inputs. By orchestrating a frozen selection mechanism for the former and a compensatory fusion mechanism for the latter, AD-BTS effectively reconciles the long-standing conflict between computational efficiency and structural integrity, thereby establishing a superior efficiency–accuracy Pareto frontier under extreme compression regimes. While our current global routing design offers a negligible-overhead solution, it also points toward a promising direction for future research: extending density-aware routing to region-level granularity.

## Impact Statement

This paper presents **AD-BTS**, a framework designed to enhance the inference efficiency of Multimodal Large Language Models (MLLMs). Our work primarily contributes to the advancement of **Green AI**. By significantly reducing the number of visual tokens required for processing high-resolution images and videos, AD-BTS lowers the computational cost (FLOPs) and memory footprint during inference. This reduction directly translates to lower energy consumption and a reduced carbon footprint for deploying large-scale multimodal systems in data centers.

Furthermore, by improving efficiency, our method facilitates the deployment of MLLMs on resource-constrained edge devices. This has the potential to democratize access to advanced AI capabilities, enabling applications such as real-time visual assistants for the visually impaired or offline document processing in regions with limited internet connectivity.

We also acknowledge potential risks. As a token sparsification technique, there is an inherent theoretical risk that aggressive pruning could lead to the loss of critical semantic details, potentially inducing hallucinations in high-stakes domains (e.g., medical imaging analysis). However, our proposed dual-branch mechanism explicitly mitigates this by preserving structural integrity in information-sparse signals. We encourage practitioners to rigorously validate the retention rate thresholds before deploying such methods in safety-critical scenarios.

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

# A. Algorithm Details and Implementation

Algorithm 1 outlines the forward pass of the proposed AD-BTS framework. The process consists of three main stages: routing, execution, and differentiable regularization.

**Preprocessing & Routing (Lines 3-5):** The pipeline begins by projecting the input image into visual tokens $\mathbf{X}$. The Gradient-based Routing Gate (GRG) then computes the structural flatness score $\mathcal{S}_{\text{flat}}$ (as derived in Appendix D). This score serves as a physical prior to classify the input as either spatially redundant input (low flatness) or a spatially sparse structured input (high flatness), determining the subsequent processing path.

**Branch Execution (Lines 7-16):**
**RSB Path (Lines 7-11):** For spatially redundant inputs, efficiency is prioritized. The visual backbone is strictly frozen to preserve pretrained feature robustness. A lightweight scorer $\phi_{\text{RSB}}$ computes token importance, and the top-$K$ tokens are gathered directly. This "select-and-freeze" strategy minimizes computational overhead.
**SFB Path (Lines 12-16):** For spatially sparse structured inputs, structural integrity is paramount. The algorithm activates Dynamic Adapter Modulation (DAM) to enable subspace adaptation. The top-$K$ tokens serve as anchors ($\mathbf{A}$) and are enhanced via a global context fusion module Eq. (6). This "Fuse-and-Adapt" strategy reconstructs the information lost during sparsification using learnable positional encodings.

**Differentiable Training (Lines 18-24):** To enable end-to-end learning under the discrete cardinality constraint $K$, we employ a continuous relaxation during training. A dynamic bias term $b^*$ is computed via a bisection solver such that the sum of the sigmoid-activated scores equals $K$. A regularization loss $\mathcal{L}_{\text{reg}}$ is then computed between this soft mask $\mathbf{m}$ and the hard binary mask $\mathbf{m}_{\text{hard}}$. Gradients for the scorer parameters propagate through $\mathcal{L}_{\text{reg}}$ via the Implicit Function Theorem (derivation in Appendix B), ensuring the selection mechanism is optimized jointly with the task objective.

---

**Algorithm 1** Adaptive Dual-Branch Token Sparsification (AD-BTS)

---

1: **Input:** Image $I$, Text $\mathbf{T}$, Budget ratio $\rho$, Flatness threshold $\delta$
2: **Output:** Sparsified Visual Tokens $\tilde{\mathbf{X}}$, Regularization Loss $\mathcal{L}_{\text{reg}}$
3: *// 1. Preprocessing & Routing*
4: $\mathbf{X} \leftarrow E_{\text{vis}}(I); \quad N \leftarrow |\mathbf{X}|; \quad K \leftarrow \lfloor \rho N \rfloor$
5: Calculate flatness score $\mathcal{S}_{\text{flat}}(I)$ via Gradient Field (Eq. 1)
6: *// 2. Branch Execution*
7: **if** $\mathcal{S}_{\text{flat}} \leq \delta$ **then**
8:     MODE $\leftarrow$ **RSB** (Natural/Dense)
9:     Freeze $E_{\text{vis}}$; Disable LoRA
10:     $\mathbf{s} \leftarrow \phi_{\text{RSB}}(\mathbf{Q}{=}\mathbf{X}, \mathbf{K}{=}[\mathbf{X}; \mathbf{T}])$ {Lightweight Scorer}
11:     idx $\leftarrow$ TopK($\mathbf{s}, K$)
12:     $\tilde{\mathbf{X}} \leftarrow$ Gather($\mathbf{X}$, idx) {Select-and-Freeze}
13: **else**
14:     MODE $\leftarrow$ **SFB** (Symbolic/Sparse)
15:     Enable LoRA (DAM) {Activate Structure Adaptation}
16:     $\mathbf{s} \leftarrow \phi_{\text{SFB}}(\mathbf{Q}{=}\mathbf{X}, \mathbf{K}{=}[\mathbf{X}; \mathbf{T}])$ {Structure Scorer}
17:     $\mathbf{A} \leftarrow$ Gather($\mathbf{X}$, TopK($\mathbf{s}, K$)) {Get Anchors}
18:     $\tilde{\mathbf{X}} \leftarrow$ Fuse($\mathbf{A}, [\mathbf{X}; \mathbf{T}], \mathbf{P}$) via Eq. 6 {Fuse-and-Adapt}
19: **end if**
20: *// 3. Differentiable Training Objective*
21: **if** is_training **then**
22:     *// Compute soft mask via implicit differentiation root-finding*
23:     $b^* \leftarrow$ BisectionSolver($\mathbf{s}, K$) {Solve $\sum \sigma(\mathbf{s} + b^*) = K$}
24:     $\mathbf{m} \leftarrow \sigma(\mathbf{s} + b^*)$
25:     $\mathbf{m}_{\text{hard}} \leftarrow$ OneHot(TopK($\mathbf{s}, K$))
26:     $\mathcal{L}_{\text{reg}} \leftarrow$ BCE($\mathbf{m}$, StopGrad($\mathbf{m}_{\text{hard}}$))
27: **else**
28:     $\mathcal{L}_{\text{reg}} \leftarrow 0$
29: **end if**
30: **Return** $\tilde{\mathbf{X}}, \mathcal{L}_{\text{reg}}$

---

## B. Derivation of Implicit Gradients for Top-K Selection

In Section 3, we employ the Implicit Function Theorem to compute the gradients of the task loss $\mathcal{L}$ with respect to the importance scores $\mathbf{s}$ through the dynamic bias $b^*$. Here, we provide the step-by-step derivation.

Recall the root-finding constraint equation at equilibrium:

$$f(b^*, \mathbf{s}) = \sum_{i=1}^{N} \sigma(s_i + b^*) - K = 0, \tag{10}$$

where $\sigma(z) = (1 + e^{-z})^{-1}$ is the sigmoid function. By the Implicit Function Theorem, the derivative of the implicit root $b^*$ with respect to the input scores $\mathbf{s}$ is given by:

$$\frac{\partial b^*}{\partial \mathbf{s}} = -\left(\frac{\partial f}{\partial b^*}\right)^{-1} \frac{\partial f}{\partial \mathbf{s}}. \tag{11}$$

First, we compute the partial derivatives of $f$. Let $z_i = s_i + b^*$. The derivative with respect to $b^*$ is:

$$\frac{\partial f}{\partial b^*} = \sum_{i=1}^{N} \sigma'(z_i) \cdot \frac{\partial(s_i + b^*)}{\partial b^*} = \sum_{i=1}^{N} \sigma'(z_i) = \|\sigma'(\mathbf{z})\|_1, \tag{12}$$

where $\sigma'(z) = \sigma(z)(1 - \sigma(z))$.

Next, the gradient with respect to the score vector $\mathbf{s}$ is:

$$\frac{\partial f}{\partial \mathbf{s}} = [\sigma'(z_1), \ldots, \sigma'(z_N)]^\top = \sigma'(\mathbf{z}). \tag{13}$$

Substituting these back gives the Jacobian of the bias term:

$$\frac{\partial b^*}{\partial \mathbf{s}} = -\frac{\sigma'(\mathbf{z})}{\|\sigma'(\mathbf{z})\|_1}. \tag{14}$$

Finally, applying the chain rule to the loss function $\mathcal{L}$. The soft mask is $\mathbf{m} = \sigma(\mathbf{s} + b^*)$. The gradient flows through both the direct path ($\mathbf{s}$) and the indirect path ($b^*$):

$$\begin{aligned}
\frac{\partial \mathcal{L}}{\partial \mathbf{s}} &= \frac{\partial \mathcal{L}}{\partial \mathbf{m}} \cdot \frac{d\mathbf{m}}{d\mathbf{s}} \\
&= \frac{\partial \mathcal{L}}{\partial \mathbf{m}} \odot \sigma'(\mathbf{z}) \cdot \left(\mathbf{I} + \frac{\partial b^*}{\partial \mathbf{s}}\right) \\
&= \frac{\partial \mathcal{L}}{\partial \mathbf{m}} \odot \sigma'(\mathbf{z}) + \left(\sum_j \frac{\partial \mathcal{L}}{\partial m_j} \sigma'(z_j)\right) \frac{\partial b^*}{\partial \mathbf{s}} \\
&= \left(\frac{\partial \mathcal{L}}{\partial \mathbf{m}} - \frac{\langle \frac{\partial \mathcal{L}}{\partial \mathbf{m}}, \sigma'(\mathbf{z}) \rangle}{\|\sigma'(\mathbf{z})\|_1}\right) \odot \sigma'(\mathbf{z}).
\end{aligned} \tag{15}$$

This matches Eq. (4) in the main text.

## C. Theoretical Complexity Analysis

We compare the asymptotic computational complexity of AD-BTS against a standard MLLM processing full visual tokens. Let $N$ be the number of visual tokens, $M$ the number of text tokens, $d$ the hidden dimension, and $K$ the retained token budget ($K \ll N$).

**Standard MLLM.** The self-attention mechanism computes pairwise interactions across all tokens. The cost for one layer is:

$$\mathcal{C}_{\text{std}} = \mathcal{O}\left((N + M)^2 d\right). \tag{16}$$

This scales quadratically with the visual sequence length $N$.

**AD-BTS Framework.** Our method introduces a scoring overhead but reduces the generation cost.

1. **Routing & Scoring (GRG + Scorer):** The GRG uses finite differences, costing $\mathcal{O}(N)$. The Scorer is a lightweight attention module. In RSB mode (self-query), it costs $\mathcal{O}(Nd^2 + N^2d)$ (if using full attention) or $\mathcal{O}(Nd^2)$ (if using linear approximation). In our implementation, we use a simplified query-key projection:

$$\mathcal{C}_{\text{score}} \approx \mathcal{O}(N \cdot d). \quad \text{(linear w.r.t } N) \tag{17}$$

2. **Generation:** After sparsification, the LLM processes only $K$ visual tokens. The complexity becomes:

$$\mathcal{C}_{\text{gen}} = \mathcal{O}\left((K + M)^2 d\right). \tag{18}$$

**Speedup Factor.** Since $K = \rho N$ where $\rho \in (0, 1)$ is the budget ratio (e.g., 0.2), the reduction in the quadratic term is significant. The theoretical speedup ratio $\gamma$ approaches:

$$\lim_{N \to \infty} \gamma = \frac{(N + M)^2}{(K + M)^2} \approx \frac{1}{\rho^2}. \tag{19}$$

For $\rho = 0.2$, this implies a theoretical $\sim 25\times$ reduction in attention computation for the visual component.

## D. Discrete Approximation of structural flatness

In Section 3, we defined the Flatness Score $\mathcal{S}_{\text{flat}}$ using the continuous gradient $\nabla I$. In our digital implementation, we approximate this using finite differences on the discrete pixel grid.

Let $I[y, x]$ denote the pixel value at coordinates $(y, x)$. The partial derivatives are approximated as:

$$\frac{\partial I}{\partial x} \approx I[y, x + 1] - I[y, x], \quad \frac{\partial I}{\partial y} \approx I[y + 1, x] - I[y, x]. \tag{20}$$

The gradient magnitude $\|\nabla I\|_1$ at position $(y, x)$ is computed as the Manhattan norm of the local differences:

$$\|\nabla I_{y,x}\|_1 \approx |I[y, x + 1] - I[y, x]| + |I[y + 1, x] - I[y, x]|. \tag{21}$$

This corresponds directly to the vectorized implementation in our GRG module:

$$\texttt{diff\_x} = |I_{:,:,1:} - I_{:,:,:-1}|, \quad \texttt{diff\_y} = |I_{:,1:,:} - I_{:,:-1,:}|. \tag{22}$$

This discrete formulation is computationally negligible ($\mathcal{O}(N)$) and fully vectorized on GPUs, ensuring that the routing step introduces no latency bottleneck.

## E. Detailed Experiment Setting

### E.1. Implementation Details

**Model Configuration.** We implement AD-BTS on top of Qwen2.5-VL-7B. The Gradient-based Routing Gate (GRG) computes the flatness score directly from the visual encoder's output gradients. The routing threshold is set to $\delta = 0.4$ across all datasets based on ablation studies (see Section 4.4). For the Dynamic Adapter Modulation (DAM), we inject Low-Rank Adapters (LoRA) into the query and value projections of the vision-language cross-attention layers. Specifically, we configure the adapters with **Rank** $r = 256$, **Alpha** $\alpha = 512$, and **Dropout** 0.05, targeting the $\texttt{q\_proj}$ and $\texttt{v\_proj}$ modules in the vision resampler. Crucially, these adapters are only active when the input is routed to the Structural Fusion Branch (SFB).

**Hardware and Environment.** All experiments are implemented in PyTorch using the Hugging Face Transformers library. Inference latency and memory profiling were conducted on a single NVIDIA H20 GPU (96GB VRAM) with a batch size of 1 to simulate real-world interactive scenarios.

## E.2. Baselines

We compare AD-BTS with the following state-of-the-art methods. For fair comparison, we utilized the official implementations and aligned the token retention rates.

**DART**(Wen et al., 2025): A training-free token pruning framework that employs $\epsilon$-Duplicate scoring to eliminate redundant visual tokens. It optimizes inference efficiency and reduces hallucinations by selecting pivot tokens that maximize information coverage, ensuring compatibility with Flash Attention mechanisms.

**Prumerge+**(Shang et al., 2024): An adaptive visual token reduction method combining sparse attention-based pruning with spectral clustering. It introduces a token supplementation phase that merges non-salient tokens into anchor representations, preserving background context while achieving substantial compression ratios in a training-free manner.

**FastV**(Chen et al., 2024): A dynamic computation reduction technique that exploits the layer-wise 'V-shape' attention pattern in MLLMs. By pruning visual tokens with low attention scores in early layers, it alleviates the computational bottleneck of deep transformer layers without requiring model retraining.

**DivPrune**(Alvar et al., 2025): A diversity-centric token pruning algorithm that formulates selection as a Max-Min Diversity optimization problem. It retains a subset of tokens that maximizes feature space coverage, effectively mitigating redundancy and preserving distinct semantic information across the visual input.

**VisionZip**(Yang et al., 2024): A text-agnostic redundancy removal method that identifies dominant tokens via self-attention clustering. It merges non-dominant tokens into their nearest dominant neighbors, significantly accelerating the pre-filling phase by compressing visual representations prior to language model interaction.

**Dynamic**(Rao et al., 2021): A class of adaptive sparsification frameworks employing learnable gating modules and Gumbel-Softmax sampling. These methods dynamically adjust token retention rates based on input complexity, optimizing the trade-off between computational efficiency and spatio-temporal representational fidelity.

**VisionSelector**(Zhu et al., 2025): An end-to-end learnable compression framework that trains a dedicated selection network to optimize token retention. It balances saliency and coverage objectives, learning to preserve semantically critical visual tokens specifically for downstream reasoning performance rather than relying on heuristic metrics.

## E.3. Datasets

### E.3.1. IMAGE UNDERSTANDING

**DocVQA**(Mathew et al., 2020): A benchmark for document understanding comprising 50,000 questions across 12,000+ document images. It evaluates OCR and reasoning capabilities using the Average Normalized Levenshtein Similarity (ANLS) metric, challenging models to interpret complex layouts, forms, and tables.

**ChartQA**(Masry et al., 2022): A visual reasoning benchmark focused on data visualizations, containing a mix of human-authored and machine-generated QA pairs. It assesses the capacity of MLLMs to perform logical and arithmetic reasoning on bar charts, line graphs, and pie charts under varied visual styles.

**TextVQA**(Singh et al., 2019): A dataset evaluating visual reasoning over text in natural images, containing 45,336 questions sourced from OpenImages. It requires models to integrate Optical Character Recognition (OCR) with scene understanding to answer questions regarding text embedded in complex real-world environments.

**OCRBench**(Liu et al., 2023): A comprehensive evaluation suite designed to assess the optical character recognition capabilities of MLLMs. It aggregates 29 datasets into five components—text recognition, scene text VQA, document VQA, KIE, and handwritten math—providing a unified metric for text-centric visual understanding.

**ScienceQA**(Lu et al., 2022): A multimodal benchmark comprising 21k multiple-choice science questions with annotated lectures and explanations. It is designed to evaluate Chain-of-Thought (CoT) reasoning capabilities in MLLMs across diverse domains including natural science, social science, and language science.

**AI2D**(Kembhavi et al., 2016): A dataset consisting of over 5,000 grade-school science diagrams with dense annotations of elements and semantic relations. It serves as a benchmark for diagram parsing and VQA, requiring models to interpret the structural topology and interactions within educational illustrations.

**MMMU**(Yue et al., 2023): A rigorous benchmark evaluating expert-level multimodal reasoning across 30 college-level subjects and six disciplines. It features heterogeneous image types and requires deep domain knowledge, challenging MLLMs to perform deliberate reasoning beyond basic visual perception.

**MME**(Fu et al., 2023) : A comprehensive benchmark assessing perception and cognition in MLLMs using a 'Yes/No' question format across 14 subtasks. It minimizes prompt engineering bias and provides fine-grained evaluation of capabilities such as object existence, counting, color recognition, and spatial reasoning.

**POPE**(Li et al., 2023b): A polling-based evaluation protocol designed to quantify object hallucination in MLLMs. It

*Table 6.* **Comparison with state-of-the-art methods across varying token retention rates.** We evaluate methods on nine benchmarks under three compression regimes (30%, 20%, and 10% retained tokens). The table reports absolute metrics for individual tasks, while the *Avg Score* column presents the performance normalized relative to the uncompressed Qwen2.5-VL-3B upper bound. **Bold** highlights the best results among compressed models.

| Method | Training Free | DocVQA (Anls) | ChartQA (Relaxed) | TextVQA (EM) | OCRBench (Score) | ScienceQA (EM) | AI2D (Acc) | MMMU (Acc) | MME (Score) | POPE (F1) | Avg Score (Norm %) |
|---|---|---|---|---|---|---|---|---|---|---|---|
| *Dynamic Resolution (MinPix=256x28x28, MaxPix=2048x28x28), Upper Bound (100%)* | | | | | | | | | | | |
| Qwen-2.5-VL-3B | – | 92.76 | 83.40 | 77.87 | 788.00 | 80.37 | 90.84 | 46.78 | 2168.46 | 86.94 | 100.00% |
| *Retain 30% Tokens (70% Compression Ratio)* | | | | | | | | | | | |
| DART | ✓ | 65.96 | 64.80 | 63.64 | 541.00 | **81.90** | 72.28 | 45.44 | 1996.75 | 84.85 | 85.27% |
| Prumerge+ | ✓ | 65.43 | 63.88 | 64.29 | 553.00 | 79.97 | 79.24 | **46.00** | 2051.01 | 84.78 | 86.34% |
| FastV | ✓ | 81.66 | 70.04 | 74.03 | 629.00 | 78.63 | 86.40 | **46.00** | 2048.53 | 82.94 | 92.01% |
| Divprune | ✓ | 73.99 | 66.48 | 70.34 | 631.00 | 80.07 | 83.42 | 45.44 | 2010.72 | **86.38** | 90.06% |
| Visionzip | ✓ | 81.38 | 75.40 | 67.71 | 635.00 | 80.02 | 84.88 | 45.44 | 2065.69 | 86.00 | 92.22% |
| Visionselector | ✗ | 88.91 | 75.28 | 75.57 | 745.00 | 79.67 | **88.73** | 45.89 | 2042.44 | 85.37 | 96.11% |
| AD-BTS (Ours) | ✗ | **89.57** | **78.44** | **76.56** | **752.00** | 80.22 | 88.60 | 45.81 | **2070.88** | 84.69 | **96.95%** |
| *Retain 20% Tokens (80% Compression Ratio)* | | | | | | | | | | | |
| DART | ✓ | 51.01 | 54.24 | 56.44 | 433.00 | **80.96** | 69.07 | 45.33 | 1925.04 | 81.93 | 78.24% |
| Prumerge+ | ✓ | 52.58 | 54.76 | 58.44 | 484.00 | 79.97 | 74.06 | 44.89 | 1977.90 | 83.23 | 80.31% |
| FastV | ✓ | 72.90 | 62.00 | 71.41 | 560.00 | 78.88 | 83.29 | 45.11 | 1992.58 | 80.02 | 87.32% |
| Divprune | ✓ | 63.82 | 57.64 | 66.76 | 535.00 | 80.47 | 77.59 | 45.56 | 1955.53 | **85.47** | 84.77% |
| Visionzip | ✓ | 69.62 | 65.76 | 59.97 | 506.00 | 79.97 | 78.79 | 45.67 | 1894.24 | 84.17 | 84.80% |
| Visionselector | ✗ | 84.03 | 72.24 | 73.34 | 684.00 | 79.57 | **86.95** | 45.33 | 2011.88 | 84.50 | 93.31% |
| AD-BTS (Ours) | ✗ | **84.23** | **75.32** | **75.18** | **689.00** | 79.97 | **86.95** | 45.78 | **2043.59** | 83.97 | **94.33%** |
| *Retain 10% Tokens (90% Compression Ratio)* | | | | | | | | | | | |
| DART | ✓ | 32.32 | 39.44 | 44.39 | 315.00 | **79.72** | 65.54 | 44.33 | 1733.77 | 75.41 | 67.99% |
| Prumerge+ | ✓ | 37.51 | 46.68 | 46.43 | 350.00 | 78.93 | 68.43 | 44.56 | 1777.61 | 77.55 | 71.16% |
| FastV | ✓ | 54.62 | 46.96 | 65.58 | 411.00 | 79.33 | 77.07 | 45.11 | 1815.70 | 70.28 | 77.35% |
| Divprune | ✓ | 47.13 | 44.36 | 57.80 | 394.00 | 78.19 | 70.14 | 44.67 | 1782.74 | 80.50 | 74.78% |
| Visionzip | ✓ | 41.67 | 46.84 | 43.83 | 321.00 | 79.47 | 69.43 | **45.22** | 1725.49 | 78.60 | 71.12% |
| Visionselector | ✗ | **67.63** | **64.96** | 63.24 | 546.00 | 79.57 | 79.27 | **45.22** | 1864.80 | **81.56** | 84.89% |
| AD-BTS (Ours) | ✗ | 67.35 | 64.76 | **68.59** | **551.00** | **79.72** | **80.44** | 44.78 | **1868.22** | 81.11 | **85.69%** |

employs random, popular, and adversarial sampling strategies to query the existence of objects, measuring the model's precision and resistance to hallucinating non-existent visual elements.

### E.3.2. VIDEO UNDERSTANDING

**MVBench**(Li et al., 2023a) : A comprehensive benchmark for evaluating temporal understanding in Video-LLMs. It comprises 20 distinct temporal tasks generated via a static-to-dynamic transformation method, assessing capabilities such as action sequence analysis, state change detection, and temporal object grounding.

**SEEDBench**(Ying et al., 2025) : A large-scale benchmark for generative multimodal comprehension, featuring 19,000 human-annotated multiple-choice questions. It covers 12 dimensions of spatial and temporal understanding, providing a stable platform for evaluating the hierarchical capabilities of image and video MLLMs.

**NextQA**(Xiao et al., 2021): A video VQA benchmark emphasizing causal and temporal action reasoning. It challenges models to answer 'why' and 'how' questions regarding object interactions in daily activities, moving beyond description to evaluate the understanding of causal chains and temporal logic in video.

## F. Additional Experiments on Qwen2.5-VL-3B

To verify the universality and scalability of our framework across different model capacities, we extend our evaluation to the lightweight Qwen2.5-VL-3B backbone. Smaller models typically possess less redundant capacity, making token sparsification significantly more challenging as every token carries higher marginal information value. We replicate the experimental protocol described in Section 4.1, maintaining identical hyperparameters ($\delta = 0.4, r = 256$) while replacing the 7B backbone with the 3B variant. We compare AD-BTS against the full spectrum of baselines across the same nine benchmarks.

Table 6 presents the comparative results. Consistent with our findings on the 7B model, AD-BTS establishes a new Pareto frontier on the 3B backbone, demonstrating robust performance retention even under rigorous parameter constraints. AD-BTS achieves the highest average score across all three compression regimes. Notably, at the aggressive 10% retention

rate, AD-BTS attains a normalized score of 85.69%, outperforming the strongest learnable baseline (VisionSelector, 84.89%) and the best training-free method (DivPrune, 74.78%) by substantial margins. This indicates that our density-aware routing effectively identifies critical information even when the backbone's intrinsic capacity is limited.

A closer inspection reveals that AD-BTS significantly excels in fine-grained textual reasoning. At 10% retention on TextVQA, AD-BTS achieves an accuracy of 68.59, surpassing VisionSelector (63.24) by 5.35 points and FastV (65.58) by 3.01 points. Similarly, on OCRBench, AD-BTS maintains a score of 551, outperforming all baselines. This confirms that the structural fusion branch is particularly effective at preserving the stroke-level details required for OCR and text recognition, preventing the semantic fracture often observed in pruning-based methods.

# G. More Visualizations

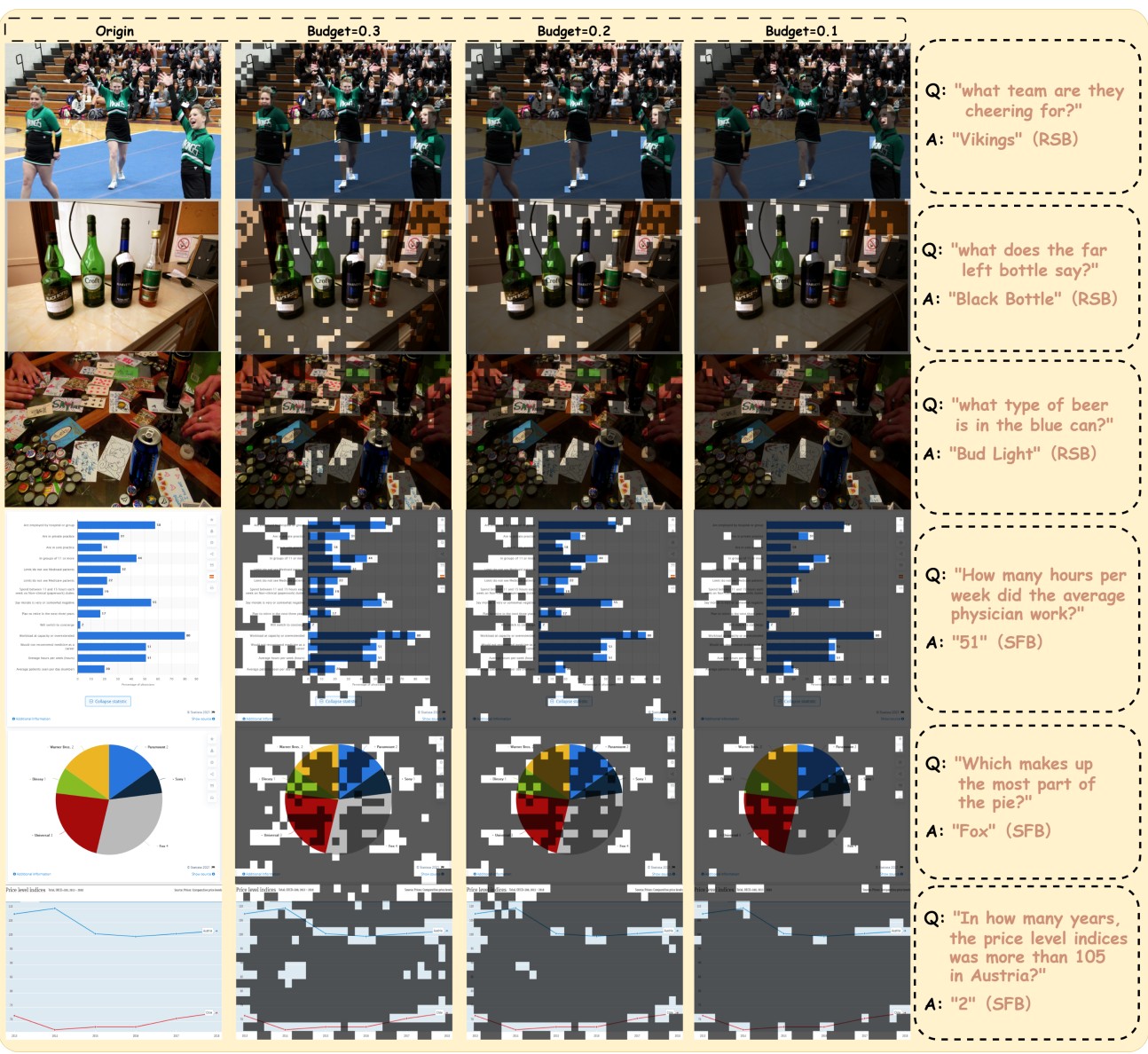

*Figure 5.* More visualization examples of AD-BTS on different budgets.

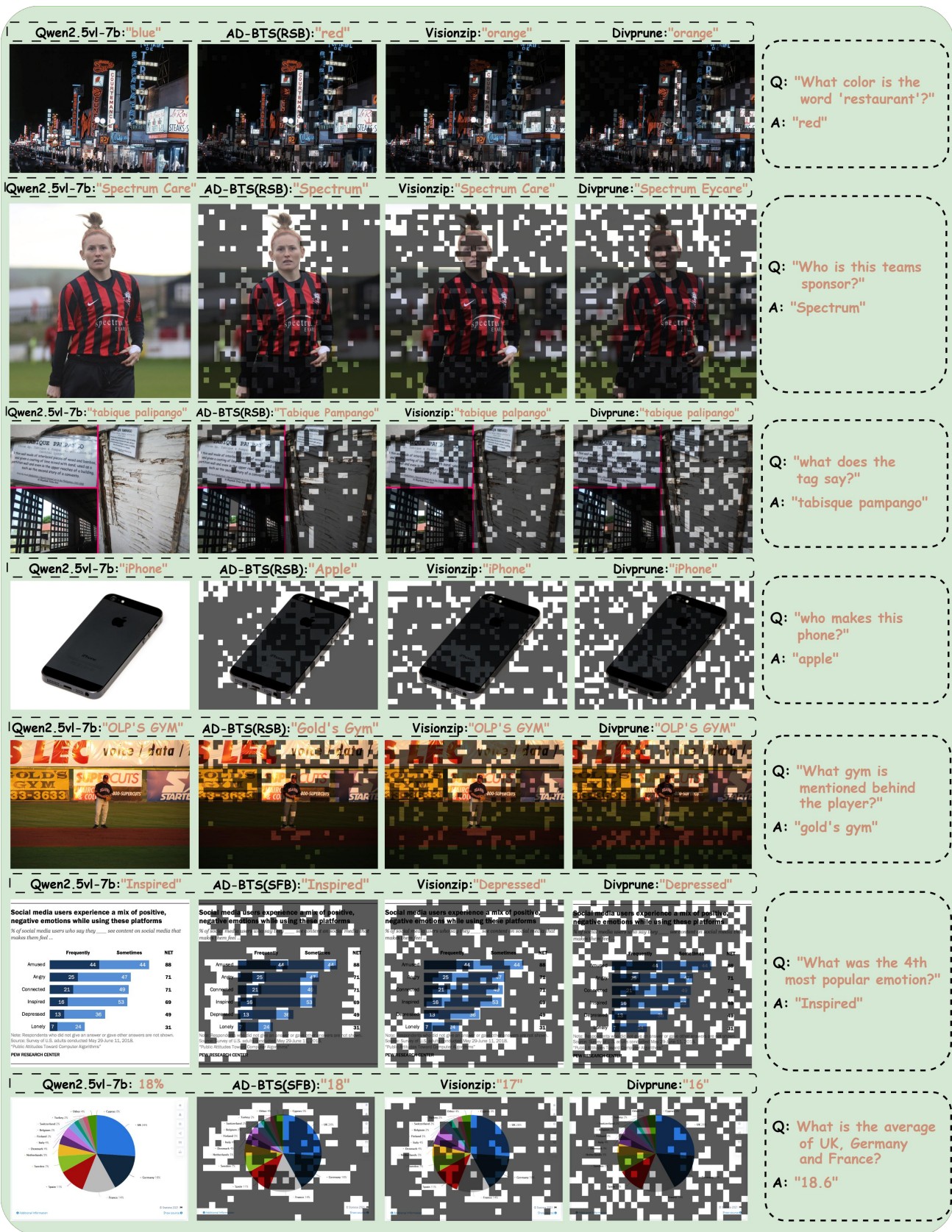

*Figure 6.* Qualitative comparison of token sparsification visualizations between AD-BTS and state-of-the-art baselines (VisionZip and DivPrune) on Qwen2.5-VL-7B.

