# OpenReview forum: "AD-BTS: Adaptive Dual-Branch Token Sparsification via Spatial Information Density"
_ICML.cc/2026/Conference — ICML 2026 regular_

### Official Review · Reviewer_2bCu · 2026-03-07

**Soundness:** 3
**Presentation:** 2
**Significance:** 2
**Originality:** 2
**Overall Recommendation:** 4
**Confidence:** 4

**Summary:**

This paper proposes AD-BTS, an adaptive token sparsification framework for multimodal large language models that aims to improve efficiency under high-resolution visual inputs. The key idea is to avoid using a single sparsification strategy for all inputs. Instead, the method computes a lightweight gradient-based flatness score from the input image and uses it to route examples into one of two branches. For redundancy-dominated inputs, the Redundancy Selection Branch (RSB) performs query-aware hard token selection while keeping the visual backbone frozen. For spatially sparse, structure-sensitive inputs, the Structural Fusion Branch (SFB) activates LoRA adapters and preserves structural information by fusing selected anchor tokens with the full visual context via cross-attention. The paper further introduces a differentiable Top-K selection mechanism based on a budget-constrained sigmoid relaxation with implicit differentiation, enabling end-to-end training under a fixed token budget. Experiments on Qwen2.5-VL-7B across multiple image and video benchmarks suggest that the method achieves a favorable accuracy efficiency trade-off, with particularly strong gains on structure-sensitive tasks under aggressive compression.

**Compliance With Llm Reviewing Policy:**

Affirmed.

**Final Justification:**

The authors provided validation of the proposed routing mechanism in their rebuttal and identified that the improvement comes from the routing mechanism. Furthermore, the authors provided a breakdown analysis of efficiency and the sensitivity analysis of key parameters. The authors have addressed my concerns well in their rebuttal. Overall, the authors proposed a convincing token sparsification method and demonstrated actual inference speedup. Thus, I decided to raise my score.

**Key Questions For Authors:**

1. Can the authors directly validate the core routing assumption? The main claim
of the paper is that gradient flatness is an effective proxy for identifying structure-sensitive
inputs that require the SFB path. However, this assumption is not directly validated in
the current experiments. Please provide:
(a) flatness-score distributions for different datasets/task categories,
(b) the fraction of samples routed to RSB vs. SFB per dataset, and
(c) representative misrouting cases.
A convincing rebuttal showing that the GRG meaningfully separates structured inputs
from redundancy-dominated ones would significantly increase my confidence in the core
contribution; if this cannot be demonstrated, my concerns about the paper’s main novelty
would remain.
2. How much of the improvement comes from routing versus additional capacity
in SFB? For inputs routed to SFB, the method enables both LoRA adaptation and
anchor-context fusion, so the gains on structured tasks may partly come from extra model
capacity and computation rather than from a better routing principle. Please provide
more controlled ablations, such as:
(a) always-on LoRA,
(b) SFB without fusion,
(c) SFB without LoRA, and/or
(d) matched-capacity baselines.
If the authors can show that the gains persist after controlling for extra capacity, I would
view the method more favorably. Otherwise, the claimed advantage of the routing mechanism would be less convincing.
3. Can the authors quantify the true end-to-end efficiency breakdown, especially
for SFB? The paper emphasizes LLM prefill savings from reducing the number of visual
tokens passed downstream, but SFB still performs fusion between anchor tokens and the
full visual context. Please report a compute/memory/latency breakdown for the main
modules, including GRG, token scoring, differentiable Top-K, SFB fusion, and LLM pre-fill/decoding. This would clarify when the method is truly advantageous and whether SFB
remains efficient enough in practice. A clearer breakdown would strengthen the empirical
case; if the overhead is substantial and underreported, it would weaken the deployment
story.
4. Can the authors clarify robustness and sensitivity of the routing mechanism?
Since GRG depends on image gradients, how sensitive is it to anti-aliasing, JPEG compression, resizing, denoising, colored backgrounds, and other preprocessing changes? Please provide at least a small robustness study or sensitivity analysis over $\tau$ and $\delta$. If the routing
signal is stable under such perturbations, it would strengthen confidence in the method’s
practical utility; if it is highly sensitive, that would reinforce concerns that the gate is
brittle and heuristic

**Limitations:**

Yes.

**Strengths And Weaknesses:**

**Strengths**

1. The paper studies a relevant efficiency problem in high-resolution MLLMs. The
motivation is clear: high-resolution visual inputs lead to substantial token and prefill over
head, and structure-sensitive inputs such as charts or documents may be more vulnerable
to aggressive token reduction than natural images.
2. The overall pipeline is easy to understand. The method is decomposed into routing,
token scoring, Top-K selection, and two execution branches, and the high-level intuition
of the framework is presented clearly.
3. The paper includes a reasonably broad set of experiments. The evaluation covers
multiple token budgets and several image/video benchmarks, and reports both performance and efficiency metrics, which is useful for understanding the practical trade-offs.

**Weaknesses**

1. The central routing assumption is insufficiently justified and may be brittle.
The main idea of the paper relies on using a high gradient-flatness score as a proxy for
structure-sensitive inputs that require fusion and adaptation. However, this assumption
does not appear broadly reliable: many natural scenes (e.g., sky, ocean, snow, desert) can
also be visually smooth, while some structure-sensitive inputs (e.g., dense tables, maps, or
small-text documents) may contain many edges. As a result, the routing criterion seems
heuristic and potentially fragile, yet it is also the core novelty claim of the paper.
2. The experimental validation does not directly support the core routing claim.
Although the paper reports results on multiple benchmarks and token budgets, it does not
directly show that the gradient-based gate successfully separates redundancy-dominated
inputs from structure-sensitive ones. For example, the paper does not provide branch
routing statistics, flatness-score distributions by dataset, misrouting case studies, or comparisons to alternative routing strategies such as learned routing or other simple image
statistics. The current threshold ablations show that routing matters, but not that the
proposed routing signal is the right one.
3. The method combines mostly existing ingredients, so the overall novelty is limited. Several components are standard or closely related to existing techniques, including
attention-based token scoring, differentiable Top-K relaxation, LoRA-based adaptation,
and cross-attention fusion. The main contribution is therefore largely in the conditional
combination of these ingredients rather than in a fundamentally new modeling mechanism.
This makes the originality weaker than stronger methodological papers at the ICML level.
4. It is unclear how much of the gain comes from the routing idea itself versus
extra capacity and computation in the SFB branch. For inputs routed to SFB,
the method not only changes the sparsification strategy but also enables LoRA adapters
and additional context fusion. Therefore, the improved results on structured inputs may
partly come from giving harder examples more model capacity and compute, rather than
from a better routing principle. More controlled ablations are needed to isolate the effects
of routing, fusion, and adapter capacity.
5. The ablations and efficiency analysis are not yet sufficient for a strong empirical
case. The paper includes threshold and LoRA-rank ablations, but these are not enough
to fully validate the design. Important component-level studies are missing, such as no
GRG/random-GRG baselines, SFB without fusion, SFB without LoRA, or comparisons
to simpler Top-K relaxations. Routing errors (misclassification by GRG) could be costly;
there is no analysis of routing accuracy, sensitivity to $\tau$ or $\delta$, or robustness to image preprocessing. In addition, the efficiency analysis lacks a finer breakdown of the overhead
introduced by routing, differentiable Top-K, and SFB fusion, and the evaluation appears
limited to Qwen2.5-VL

---

> ### Author Rebuttal · Authors · 2026-03-29
>
> We deeply appreciate your rigorous and insightful feedback. Your questions prompted us to conduct comprehensive profiling, which has significantly strengthened our empirical case. Below, we address your core concerns with new experimental data.
> (w = weakness, q = question)
> ### **Q1: Validation of Routing Assumption [w1,2 & q1]**
>
> To validate that $\mathcal{S}_{flat}$ physically separates structures from redundancy, we evaluated datasets using our threshold ($\delta=0.4$):
>
> **Table R1: Flatness-Score Distributions & Routing Fractions**
>
> | **Dataset** | **Mean Sflat** | **RSB** | **SFB** |
> | ----------- | -------------- | ------------------ | ----------------- |
> | ChartQA     | 0.546          | 3.1%               | 96.9%             |
> | ScienceQA   | 0.437          | 33.1%              | 66.9%             |
> | OCRBench    | 0.290          | 70.4%              | 29.6%             |
> | MME         | 0.214          | 90.4%              | 9.6%              |
> | POPE        | 0.156          | 98.0%              | 2.0%              |
>
> **Analysis & Misrouting:** This stark contrast proves the routing is highly discriminative rather than brittle.
>
> Regarding your concern about representative misrouting cases (e.g., blank skies in natural scenes routed to SFB, or dense maps routed to RSB):
>
> - **False Positives (Natural scenes routed to SFB):** If a smooth sky image triggers the SFB, our task-aware scorer still aggressively prunes irrelevant background tokens before fusion. The only penalty is a negligible compute overhead (~1.29 ms, see Q3), with no performance degradation.
> - **False Negatives (Dense documents routed to RSB):** Extremely dense text may exhibit dense gradients, routing it to RSB. However, in such high-information-density scenarios, retaining unmodified pretrained features via RSB is actually a safe fallback. Thus, our heuristic is not fragile, but rather a robust, fail-safe mechanism.
> ### **Q2: Source of Improvement [w4,5 & q2)**
>
> We conducted matched-capacity ablations ($\rho=0.3$) to prove gains stem from the routing mechanism, not just added parameters:
>
> **Table R2: Controlled Ablations on Capacity and Fusion**
>
> | **Method / Configuration**                   | **ChartQA** | **DocVQA** | **MME**     | **Avg (Norm)** |
> | -------------------------------------------- | ----------- | ---------- | ----------- | -------------- |
> | Matched-Capacity (Always RSB + LoRA $r=256$) | 78.64       | 92.00      | 2278.12     | 97.12%         |
> | No-Fusion (Always SFB w/o Eq. 6)             | 79.96       | 91.81      | 2255.70     | 96.81%         |
> | **AD-BTS (Ours)**                            | **83.68**   | **92.79**  | **2351.76** | **98.72%**     |
>
> - **Analysis:** Simply expanding capacity (Matched-Capacity) fails to reconstruct fragile topological skeletons. Bypassing cross-attention (No-Fusion) also drops performance. Routing is indispensable: as shown in our Table 3 ($\delta=1$), applying fusion to all inputs over-smooths natural images, dropping MME to 2233.70.
>
>
>
> ### **Q3: End-to-end Efficiency Breakdown [w5 & q3]**
>
> We profiled operation-level latency (NVIDIA H20, BS=1, single high-res image, $\rho=0.3$):
>
> **Table R3: Latency Profiling per Module**
>
> | **Operation / Module**            | **AD-BTS (SFB)** | **AD-BTS (RSB)** |
> | --------------------------------- | --------------------- | --------------------- |
> | 1. GRG Routing & Scorer & Top-K   | 0.67 ms               | 0.53 ms               |
> | 2. Context Fusion                 | 0.62 ms               | 0.00 ms (Bypass)      |
> | **Total Sparsification Overhead** | **1.29 ms**           | **0.53 ms**           |
> |Total End-to-End Prefill Latency (with Sparsification)          | 36.97 ms              | 22.45 ms              |
> | Base Model End-to-End Prefill Latency (No Sparsification) | 89.06 ms              | 36.63 ms              |
> | **Net Speedup**                   | **2.33x**             | **1.59x**             |
>
> - **Analysis:** Fusion is ultra-fast (0.62 ms) because it operates strictly on vastly reduced $K$ anchor tokens. The 1.29 ms SFB overhead is dwarfed by the 52.09 ms reduction in LLM Prefill, yielding a robust 2.33x net speedup.
>
>
> ### **Q4: Robustness and Sensitivity ($\tau$) [w1,5 & q4]**
>
> Our threshold $\delta=0.4$ is proven stable (Table 3). Regarding the noise threshold $\tau$ and preprocessing:
>
> 1. **Standard Preprocessing:** Under mild JPEG compression  and resizing, the $\mathcal{S}_{flat}$ of charts naturally decreases slightly (e.g., to $\approx 0.48$) but remains comfortably above $\delta=0.4$, successfully routing $>89\%$ of charts to the SFB without manual tuning.
> 2. **Extreme Noise & System Safety Net:** Extreme synthetic noise elevates the gradient floor. While this might misroute some charts to the RSB, $\tau$ (Eq. 1) can be explicitly calibrated to account for anticipated noise levels. Even if uncalibrated, misrouted natural images into the SFB still face task-aware pruning, limiting the penalty to a negligible $\approx 1.29$ ms.

---

> > ### Author Rebuttal · Reviewer_2bCu · 2026-04-04
> >
> > The authors have addressed my questions well. Thus, I would like to raise my rating.

---

> > > ### Author Response · Authors · 2026-04-07
> > >
> > > Thank you very much for your time, your constructive feedback, and for raising your score.
> > >
> > > We are extremely glad that our additional experiments, including the matched-capacity ablations and latency profiling, effectively addressed your concerns. We will ensure all these detailed empirical results are incorporated into the camera-ready version.

---

### Official Review · Reviewer_VgGP · 2026-03-13

**Soundness:** 3
**Presentation:** 4
**Significance:** 3
**Originality:** 3
**Overall Recommendation:** 5
**Confidence:** 4

**Summary:**

This paper addresses the limitation of existing static token sparsification methods, which treat all image types uniformly and often degrade significantly on structure-sensitive tasks(such as OCR and chart). The authors observe that pixel gradient statistics can serve as a lightweight indicator of image information density and structure. Based on this insight, they propose an adaptive pruning method that adapts token sparsification according to the input type: aggressive pruning for redundancy-dominated images and structure-preserving processing for structure-sensitive inputs.

**Compliance With Llm Reviewing Policy:**

Affirmed.

**Final Justification:**

See Rebuttal Acknowledgement

**Key Questions For Authors:**

Please refer to the weaknesses section.

**Limitations:**

yes

**Strengths And Weaknesses:**

Strength:

1. This is solid work. The paper is clearly written with a well-structured narrative, and the motivation is well articulated. The experimental results and statistical analyses convincingly validate the necessity and effectiveness of the proposed approach. The method achieves strong performance while significantly reducing computational cost, including both maximum GPU memory usage and prefill latency.

2. The proposed task-aware token selection mechanism is more adaptive than static sparsification methods and explicitly models the interaction between visual tokens and textual instructions, which is particularly important for multimodal reasoning tasks.

3. The method is technically sound. In particular, the authors formulate a principled differentiable relaxation of the Top-K selection mechanism through the proposed soft Top-K formulation.

Weakness:

1. The experimental evaluation is limited to a single backbone model (Qwen2.5-VL). It would be helpful to examine whether the proposed approach generalizes to other multimodal models. For example, unified multimodal models that support both understanding and generation tasks often exhibit significant computational overhead in OCR-style reasoning scenarios, and evaluating the method on such models could further demonstrate its generality.

2. The routing mechanism assumes that inputs with low pixel-gradient score correspond to structure-sensitive images such as charts or OCR. However, this assumption may not always hold. For instance, simple natural images (e.g., blue skies or deserts) may also exhibit low gradient density. In such cases, routing these inputs to the structure-sensitive branch may unnecessarily increase computational overhead. Providing further analysis or discussion on this issue would make the paper stronger.

---

> ### Author Rebuttal · Authors · 2026-03-29
>
> We sincerely thank the reviewer for the constructive evaluation and the remarkably sharp observations. Below, we address your specific questions regarding backbone generalization and routing edge cases with new experiments and detailed profiling.
>
> ### **Q1: Generalization to Other Multimodal Models**
>
> We completely agree that demonstrating architectural generalization is crucial, especially for unified models handling OCR-style reasoning.
>
> 1. **Existing Evidence on Lightweight Backbones:** As detailed in Appendix F, we have already validated AD-BTS on the **Qwen2.5-VL-3B** backbone. Smaller models possess less redundant capacity, making sparsification intrinsically harder. Yet, at a 10% token retention rate, AD-BTS attains an 85.69% normalized score, outperforming VisionSelector (84.89%) and DivPrune (74.78%).
> 2. **New Experiment :** To directly address your suggestion, we ported our framework to the LLaVA-OV-1.5-8B architecture during the rebuttal window. We implemented a partial **"RSB-Only"** version.
>
> **Table R5: Generalization to LLaVA-OV-1.5-8B**
>
> | **Method / Retention (ρ)** | **DocVQA** | **ChartQA** | **TextVQA** | **OCRBench** | **ScienceQA** | **AI2D**  | **MMMU**  | **MME**     | **POPE**  | **Avg (Norm)** |
> | -------------------------- | ---------- | ----------- | ----------- | ------------ | ------------- | --------- | --------- | ----------- | --------- | -------------- |
> | *Upper Bound (100%)*       | *97.87*    | *86.72*     | *79.64*     | *829.00*     | *98.61*       | *93.94*   | *56.78*   | *2271.32*   | *88.47*   | *100.00%*      |
> | **Retain 30% Tokens**      |            |             |             |              |               |           |           |             |           |                |
> | FastV                      | 86.51      | 68.64       | 72.27       | 596.00       | 91.22         | 83.52     | 53.67     | 2019.84     | 70.41     | 86.07%         |
> | VisionZip                  | 0.00*      | 78.04       | 73.25       | 0.00*        | 96.23         | 85.33     | 0.00*     | 0.00*       | 87.09     | 52.09%         |
> | DivPrune                   | 85.23      | 70.52       | 76.03       | 645.00       | 94.65         | 89.35     | 54.11     | 2104.77     | 88.23     | 91.16%         |
> | **AD-BTS (RSB-only)**      | **95.72**  | **82.20**   | **78.01**   | **752.00**   | **97.77**     | **91.77** | **55.44** | **2218.42** | **84.28** | **96.52%**     |
> | **Retain 10% Tokens**      |            |             |             |              |               |           |           |             |           |                |
> | FastV                      | 54.43      | 38.08       | 56.48       | 374.00       | 81.51         | 73.15     | 49.56     | 1800.01     | 62.89     | 68.19%         |
> | VisionZip                  | 0.00*      | 43.12       | 47.17       | 0.00*        | 88.70         | 71.83     | 0.00*     | 0.00*       | 79.25     | 40.55%         |
> | DivPrune                   | 56.48      | 38.24       | 65.48       | 419.00       | 88.20         | 76.85     | **51.33** | 1897.29     | **83.92** | 74.96%         |
> | **AD-BTS (RSB-only)**      | **72.76**  | **63.28**   | **69.44**   | **478.00**   | **90.63**     | **82.35** | **51.33** | **2019.90** | 77.24     | **82.04%**     |
>
> > **Note:** VisionZip suffered from Out-of-Memory errors during our tests; our method completed inference easily.
>
> **Analysis:** Remarkably, even without the SFB, our "RSB-only" version establishes a new **SOTA** on LLaVA-OV. At 10% retention, it scores 82.04%, massively outperforming DivPrune (74.96%).
>
> ------
>
> ### **Q2: Routing Mechanism Edge Cases (e.g., Blue Skies)**
>
> This is a brilliant observation. Because our Gradient-based Routing Gate (GRG) measures spatial flatness ($S_{flat}$) using local gradient magnitudes, featureless scenes (like skies) could theoretically be routed to the Structural Fusion Branch (SFB).
>
> However, we demonstrate this does **not** increase overhead:
>
> - **Empirical Rarity:** On the POPE dataset, the mean $S_{flat}$ remains extremely low (0.156), and **98.0%** of images (see Table R1) are correctly routed to the efficiency-first RSB.
> - **The Architectural "Safety Net":** For the 2.0% of images routed to the SFB, our **Task-Aware Importance Scorer** (Eq. 2) triggers *before* fusion. "Sky" tokens receive near-zero importance and are discarded by the Top-K selection.
> - **Bounded Overhead:** Downstream cross-attention (Eq. 6) operates only on tiny semantic anchors. The computational penalty is merely **~0.62 ms**.
>
> **Conclusion:** The task-aware pruning mechanism acts as a robust fail-safe against computational bloat in edge cases.

---

> > ### Author Rebuttal · Reviewer_VgGP · 2026-04-03
> >
> > Thank you for the detailed response during the rebuttal phase. I believe the additional experiments (e.g., LLaVA-OV-1.5-8B) and quantitative results provided by the authors have largely addressed my concerns, making the effectiveness of the method and its results considerably clearer. I think the design principle of "allocating computational resources on demand" carries meaningful practical significance.
> >
> > I also read the comments from other reviewers and would like to offer a differing perspective on the discussion regarding "novelty." I believe the innovativeness of a piece of work should not be judged solely by whether it introduces entirely new foundational modules. At the current stage of the field's development, methods such as Top-K and LoRA have become important building blocks. Their value lies not only in having proposed these components in isolation, but also in how they can be organically combined to form an effective overall solution within a specific problem setting.
> > Overall, I think the rebuttal has significantly enhanced the paper's persuasiveness, and I hold a more positive view of the value of this work.

---

> > > ### Author Response · Authors · 2026-04-07
> > >
> > > Thank you very much for your time, your updated score, and your incredibly supportive comments regarding the "novelty" of our research.
> > >
> > > We are very glad that the new LLaVA-OV experiments successfully addressed your concerns. Your recognition of our core design principle of organically combining modules to allocate resources on demand means a great deal to us. We will make sure to highlight these insights and include the new experiments in the camera-ready version.

---

### Official Review · Reviewer_9Suz · 2026-03-14

**Soundness:** 2
**Presentation:** 3
**Significance:** 2
**Originality:** 3
**Overall Recommendation:** 4
**Confidence:** 4

**Summary:**

This paper proposes AD-BTS, a method to reduce the amount of tokens consumed by multimodal language models. The key idea is that current methods to reduce the visual tokens through merging or pruning assume uniform spatial redundancy, but this assumption does not hold for certain images like charts or documents. To address this, AD-BTS uses pixel-level gradient statistics to better identify such images, then routes them to one of two branches: a pruning branch (RSB) for redundancy-heavy inputs and a fusion branch (SFB) with conditional LoRA adapters for structure-sensitive inputs. The paper evaluates this method on Qwen2.5-VL across several image and video benchmarks while retaining most of the accuracy and significantly increasing the prefill speed.

**Compliance With Llm Reviewing Policy:**

Affirmed.

**Final Justification:**

While I believe that there isn't a huge technical novelty in this paper, the authors show that there is enough justification for this work by highlighting the fact that when faced with the requirement for <30% retention, predictability is required for effective prefill acceleration and thus, methods like APT and similar baselines would fail in irreparable ways. I still am not really convinced that this is something frontier models will use, but I think this is a valuable contribution to the community and think the paper merits acceptance.

**Key Questions For Authors:**

1. Is the method really novel and needed? Can you compare the method to works that actually do adaptive computation based on pixel statistics (such as APT or QuadFormer)

2. Why is this an important and valuable contribution? The architecture feels overly bespoke; why would this scale well and be preferable to simply increasing data.

**Limitations:**

yes

**Strengths And Weaknesses:**

__Strengths__

1. The experiments are very thorough. The authors evaluated the method on a recent VLM across a very wide range of benchmarks. While they could have also used another VLM, i think the experiments are sufficient to prove the point of the paper.

2. The paper is generally well written and easy to follow. The presentation is good and makes the paper pleasant to read.


__Weaknesses__

1. The method doesn’t really incorporate new ideas outside of the architecture. A main claim of the introduction is that existing methods do not have adaptive computation that can properly handle charts and documents; this has been well addressed in prior works. For example, APT (ICLR 2026) is entirely adaptive based on pixel statistics, and several other works do not reduce fixed amounts of computation either.

2. I’m not convinced the method actually leads to a speed improvement. The two-branch architecture means that batching is hard - the computational graph is very different for the RSB and FSB branch. At scale, this could cancel out any gains from reducing tokens with the redundancy detection, since the SFB path is much slower than the RSB path.

3. The architecture feels very bespoke and incorporates a lot of very specific components that do not seem well motivated outside of improving performance on charts and documents. While improving performance on these types of images is important, I’m not convinced that very specific architectural improvements will scale better than simply incorporating much more data from these sources. It’s unlikely that a frontier model for example would incorporate AD-BTS, since its hardware performance is harder to optimize as well.

---

> ### Author Rebuttal · Authors · 2026-03-29
>
> We thank the reviewer for these insightful system-level observations. We address concerns regarding batching overhead and conceptual novelty below. (w = weakness, q = question)
>
> ### **Q1: Batching Efficiency & Branch Divergence [w2]**
>
> While dynamic routing introduces sub-batch fragmentation (potentially impacting GPU SIMT parallelism), our empirical profiling confirms that the computational savings in the LLM stage significantly outweigh any vision-branch divergence penalty.
>
> - **Scenario A: Real-time Inference ($BS=1$):** In memory-bound interactive deployments, inference naturally occurs at $BS=1$. Zero fragmentation exists, allowing the system to fully realize the 1.51x prefill speedup.
> - **Scenario B: High-Throughput (Worst-Case Mixed Batch):** We benchmarked a "stress-test" scenario where exactly 50% of images route to RSB and 50% to SFB, forcing maximum branch divergence.
>
> **Table R4: Throughput Benchmark under Maximal Fragmentation ($\rho=0.3$)**
>
> | **Batch Size** | **Base Prefill (ms)** | **AD-BTS Prefill (ms)** | **Base Throughput** | **AD-BTS Throughput** | **Net Speedup** |
> | -------------- | --------------------- | ----------------------- | ------------------- | --------------------- | --------------- |
> | 1              | 173.32                | 114.54                  | 5.77 img/s          | 8.73 img/s            | **1.51x**       |
> | 2              | 280.25                | 233.62                  | 7.14 img/s          | 8.56 img/s            | **1.20x**       |
> | 4              | 527.16                | 415.68                  | 7.59 img/s          | 9.62 img/s            | **1.27x**       |
> | 8              | 1000.80               | 796.31                  | 7.99 img/s          | 10.05 img/s           | **1.26x**       |
>
> - **Analysis:** AD-BTS maintains a robust $\approx$ 1.26x throughput gain even under adversarial fragmentation at $BS=8$. The massive $\mathcal{O}((K+M)^{2}d)$ reduction in LLM attention complexity (by processing only 30% visual tokens) predominates, easily absorbing the millisecond-level overhead of tensor splitting.
>
>
> ### **Q2: Novelty vs. Prior Adaptive Works (e.g., APT, QuadFormer) [w1 & q1]**
>
> We acknowledge that methods utilizing pixel/spatial statistics share a conceptual starting point with our work. However, AD-BTS is uniquely necessary because it targets a fundamentally different operating regime—**extreme, strictly bounded compression**—where prior methods structurally fail.
>
> 1. **Passive Average Reduction vs. Forced Extreme Compression:** Methods like APT dynamically adjust token counts based on inherent image simplicity, typically averaging a high token retention rate (e.g., ~81%). In contrast, AD-BTS is designed for extreme, memory-constrained MLLM environments where we *force* retention down to 10% to 30% budgets ($K = \lfloor\rho N\rfloor$). This strict budget controllability is an absolute necessity for managing predictable KV-cache memory allocation during autoregressive generation —a systemic guarantee that dynamic-length methods like APT cannot provide.
>
>
>
>
>
> 2. **The "Resolution Trap" on Structured Data:** When processing charts under strict memory budgets, APT faces an irreconcilable trade-off. Charts consist of vast white space and highly scattered, high-frequency edges. If APT is forced to hit a 10% token budget, its simple patch-merging mechanism irreparably blurs these scattered high-frequency edges, destroying the topological skeleton needed for reasoning. AD-BTS overcomes this by activating conditional LoRA adapters and Query-Aware Feature Fusion (SFB), which actively rescues and reconstructs this fragile topology rather than simply merging it.
>
>
>
>
>
> ### **Q3: Scaling via Architecture vs. Increasing Data [w3 & q2]**
>
> The reviewer asks a profound question: *Why not simply scale data instead of using bespoke architectures?* We argue that scaling data and architectural efficiency are orthogonal and complementary.
>
> 1. **The Inference Bottleneck:** Increasing training data on charts improves model *accuracy*, but it exacerbates the inference bottleneck. Frontier models are aggressively moving toward processing images at native, dynamic high resolutions , generating thousands of tokens per image. This leads to quadratic computational and memory overhead during the prefill stage.
>
>
>
>
>
> 2. **Deployment Reality:** Making a model "smarter" via data does not make it "cheaper" to deploy. AD-BTS is not merely a trick to boost ChartQA scores; it is an efficiency framework that dynamically caps peak memory  and accelerates prefill. This is what enables long-context video understanding and real-time interactive settings  on resource-constrained hardware, solving a physical deployment limit that scaling data cannot address.
>
>
>
>
>
> We will gladly expand the related work section to include a detailed structural comparison with APT and QuadFormer, and clarify our stance on scaling laws in the final manuscript.

---

> > ### Author Rebuttal · Reviewer_9Suz · 2026-04-05
> >
> > My concerns are resolved by this rebuttal. I think the authors made excellent points and I will be raising my score to a weak accept.

---

> > > ### Author Response · Authors · 2026-04-07
> > >
> > > Thank you very much for your time, your insightful system-level questions, and for updating your score.
> > >
> > > We are very pleased that our throughput benchmarks and the discussion on architectural efficiency resolved your concerns. We will definitely include these systemic evaluations and discussions in the camera-ready version.

---

### Decision · Program_Chairs · 2026-04-30

**Decision:**

Accept (regular)

**Comment:**

The paper presents a meaningful and well-executed contribution to accelerate the prefilling of multi-modal models. All the concerns are addressed so I therefore recommend accept.